# Learning Gaussian Graphical Models from a Glauber Trajectory Without Mixing

Eric Shen [* 1]   Tony Wu [* 1]   Mahbod Majid [1]   Ankur Moitra [1]

## Abstract

We study the task of learning the structure of a $d$-sparse Gaussian graphical model on $n$ variables from a single trajectory of Glauber dynamics. Beyond algorithmic considerations, many applications present temporally correlated observations rather than i.i.d. samples. In the classical i.i.d. setting, under comparably general sparsity and minimum edge-strength assumptions, sublinear-in-$n$ sample guarantees are known, but achieving them in polynomial-time remains open. Motivated in part by this gap, we give a polynomial-time algorithm that recovers the conditional-independence graph from a single Glauber trajectory, with a trajectory-length guarantee that does not depend on the mixing time.

Technically, our algorithm has three components. First, we estimate the conditional variances and rescale the trajectory to reduce to the unit-diagonal case, without changing the underlying graph. Second, we design a local edge test that extracts adjacency information from short update windows by isolating pairwise influence. Third, we aggregate these local statistics using a robust median-based estimator, and prove accuracy despite temporal dependence arising from a single trajectory.

## 1. Introduction

A *Gaussian Graphical Model* (GGM) on $n$ vertices is a mean-zero Gaussian random variable $X \sim \mathcal{N}(0, \Sigma)$. The relevant object for the graph structure is the *precision matrix* $\Theta := \Sigma^{-1}$. We associate to $\Theta$ an undirected graph $G = (V, E)$ by putting an edge $(i, j)$ whenever $\Theta_{ij} \neq 0$. The key fact is that, for Gaussians, zeros in $\Theta$ exactly encode

---

[1]Mathematics Department, Massachusetts Institute of Technology, Cambridge, MA, USA. Correspondence to: Eric Shen <eys@mit.edu>, Tony Wu <tonyyzwu@mit.edu>.

*Proceedings of the 43rd International Conference on Machine Learning*, Seoul, South Korea. PMLR 306, 2026. Copyright 2026 by the author(s).

conditional independences: for distinct $i, j \in V$,

$$X_i \perp X_j \mid X_{V \setminus \{i,j\}} \quad \Longleftrightarrow \quad \Theta_{ij} = 0.$$

This is known as the Markov property. An important measure of complexity of GGMs is *sparsity*: we say the GGM is $d$-sparse if every vertex has at most $d$ neighbors in $G$, equivalently each row of $\Theta$ has at most $d$ nonzero off-diagonal entries.

GGMs provide a natural way to represent *conditional dependence structure* among many interacting variables. The literature on GGM applications is too vast to survey here, but representative examples include neuroscience and brain connectivity (Dyrba et al., 2020; Huang et al., 2010), genomics (Yi et al., 2022), metabolic pathway reconstruction (Krumsiek et al., 2011), climate science (Zerenner et al., 2014), financial systemic-risk modeling (Cerchiello & Giudici, 2016), and environmental psychology (Bhushan et al., 2019). A recurring regime in such applications is high dimensionality, where the number of variables $n$ can be comparable to or larger than the number of available observations, motivating our focus on sparse GGMs, in which the conditional-independence graph has maximum degree at most $d$.

Algorithmically, the main challenge is already present in *structure learning*: recovering the edge set $E$ (equivalently, the support of $\Theta$). Indeed, once $G$ is known, estimating the coefficients reduces to running $n$ low-dimensional (regression) problems, each involving only the $d$ neighbors of a node. More precisely, for each node $i \in V$, the conditional distribution of $X_i$ given the remaining coordinates is

$$X_i = -\sum_{j \in N(i)} \frac{\Theta_{ij}}{\Theta_{ii}} X_j + \xi_i, \quad \xi_i \sim \mathcal{N}\left(0, \frac{1}{\Theta_{ii}}\right).$$

Thus, given $G$, estimating $\Theta$ reduces to $n$ regressions of $X_i$ onto $\{X_j : j \in N(i)\}$, which recover the coefficients $\{-\Theta_{ij}/\Theta_{ii}\}_{j \in N(i)}$ and the noise variance $1/\Theta_{ii}$.

In the classical i.i.d. data model, Misra, Vuffray and Lokhov (Misra et al., 2020) studied the information-theoretic sample complexity of learning sparse GGMs *without* assuming bounded spectrum or incoherence. The only assumption

they make is the following guarantee on the minimum normalized edge strength

$$\frac{|\Theta_{ij}|}{\sqrt{\Theta_{ii}\Theta_{jj}}} \geq \alpha \qquad \forall (i,j) \in E, \qquad \text{(non-degeneracy)}$$

which is a natural non-degeneracy condition ensuring that present edges are not arbitrarily weak. It is important to note that this constraint does *not* impose any assumptions on the minimum eigenvalue of the normalized matrix, and the spectrum may be arbitrarily ill-conditioned. For a simple demonstration of this, see appendix Section I.

They show that information-theoretically $O(d \log n/\alpha^2)$ i.i.d samples suffice for learning the graph structure. Earlier work of Wang, Wainwright, and Ramchandran (Wang et al., 2010) shows that at least $\Omega(\log n/\alpha^2)$ i.i.d samples are necessary for this task, and it is currently unknown which of the upper bound or the lower bound is tight. However, the price paid is computational: the algorithm of (Misra et al., 2020) uses an exhaustive search based on an $\ell_0$-constrained sparse linear regression and runs in time $n^{\Omega(d)}$. Whether one can match the information-theoretic sample complexity with a polynomial-time algorithm for general GGMs remains open. More broadly, there is evidence for computational barriers in related sparse linear regression problems. In particular, in the fixed-design, worst-case setting, *proper* sparse linear regression—meaning the algorithm must output a $k$-sparse predictor when a $k$-sparse solution exists—is NP-hard (Natarajan, 1995; Zhang et al., 2014).

In many scientific settings, observations are not i.i.d.; instead we observe a system evolving over time. A natural stylized model for such temporal dependence is a Markov chain whose stationary distribution is a GGM, for instance single-site Gibbs sampling (Glauber dynamics). If the chain mixes rapidly, then by spacing observations by at least the mixing time one can obtain approximately independent draws and reduce to the classical i.i.d. setting.

However, mixing-based reductions can be unsatisfactory even for Gaussian targets. Indeed, for a multivariate normal target, single-site Gibbs has an explicit linear-operator description, and its convergence rate is controlled by the spectrum of an associated update matrix (Amit, 1991; Roberts & Sahu, 1997). Moreover, this convergence behavior is invariant under diagonal rescaling of the coordinates, so it is expressed in terms of the normalized precision matrix $\Theta' = D^{-1/2}\Theta D^{-1/2}$, where $D = \text{diag}(\Theta_{11}, \ldots, \Theta_{nn})$. For example, a standard spectral-gap bound for single-site Gibbs implies

$$t_{\text{mix}}(\varepsilon) \approx n\,\frac{1}{\lambda_{\min}(\Theta')}\,\log(1/\varepsilon), \qquad (1)$$

where the equality is up to absolute constants (Roberts & Sahu, 1997; Amit, 1991). Since $\lambda_{\min}(\Theta')$ can be arbitrarily

small under the non-degeneracy constraint,[1] the mixing time can be arbitrarily large.

This leads to our main question:

> *Is it possible to learn the structure of a sparse GGM from a single Glauber trajectory* without *waiting for the chain to mix, and without imposing additional assumptions on the precision matrix $\Theta$?*

We answer this question in the affirmative by giving an efficient algorithm that recovers the graph from a single trajectory, with no dependence on the mixing time and without imposing additional assumptions on the precision matrix beyond the non-degeneracy condition. In the next section we formalize the model and state our main theorem.

### 1.1. Results

We begin by recalling two definitions that formalize our setting: $(\alpha, d)$-sparse Gaussian graphical models and single-site Glauber dynamics.

**Definition 1.1** ($(\alpha, d)$-sparse Gaussian graphical model)**.** Let $\Sigma \in \mathbb{R}^{n \times n}$ be positive definite and let $\Theta := \Sigma^{-1}$. For $\alpha > 0$ and an integer $d \geq 1$, we say that $X \sim \mathcal{N}(0, \Sigma)$ is an $(\alpha, d)$-sparse GGM if the graph $G = (V, E)$ on $V = [n]$ defined by

$$(i,j) \in E \quad \Longleftrightarrow \quad \Theta_{ij} \neq 0,$$

has maximum degree at most $d$, and moreover every edge has normalized strength at least $\alpha$, i.e.,

$$\left|\frac{\Theta_{ij}}{\sqrt{\Theta_{ii}\Theta_{jj}}}\right| \geq \alpha \qquad \forall (i,j) \in E.$$

We refer to $G$ as the (conditional-independence) graph or the sparsity pattern of the model.

Next we define the Glauber dynamics for a GGM. We work with the continuous-time dynamics, where each coordinate has an independent rate-1 Poisson clock, so the chain performs $n$ updates per unit time in expectation.

**Definition 1.2** (Continuous-time Glauber dynamics for a GGM)**.** The continuous-time Glauber dynamics for a GGM with precision matrix $\Theta$ is a random process $\{Y^{(t)}\}_{t \geq 0}$ taking values in $\mathbb{R}^n$. It is initialized at an arbitrary (possibly random or worst-case) vector $Y^{(0)}$, and is updated at random times $\{S^{(\ell)}\}_{\ell \in \mathbb{N}}$ with $S^{(0)} = 0$, where $S^{(\ell)}$ is the time of the $\ell$-th update. The inter-update times $\{S^{(\ell+1)} - S^{(\ell)}\}_{\ell \geq 0}$ are i.i.d. sampled from the exponential distribution with parameter $n$, namely $\text{Exp}(n)$, so the chain performs $n$ updates

---

[1]See Section I for a simple demonstration of this.

per unit time in expectation. The process is piecewise constant between updates; define the embedded discrete-time chain $X^{(\ell)} := Y^{(t)}$ for $t \in [S^{(\ell)}, S^{(\ell+1)})$.

At each update time $S^{(\ell)}$, an index $I^{(\ell)} \in [n]$ is chosen uniformly at random. Let $i = I^{(\ell)}$. Then we resample coordinate $i$ from its conditional distribution given the others:

$$X_i^{(\ell)} \sim \mathcal{N}\left(-\sum_{j \in N(i)} \frac{\Theta_{ij}}{\Theta_{ii}} X_j^{(\ell-1)}, \frac{1}{\Theta_{ii}}\right),$$

and set $X_j^{(\ell)} = X_j^{(\ell-1)}$ for all $j \neq i$. Here $N(i) = \{j \neq i : \Theta_{ij} \neq 0\}$ denotes the neighborhood of $i$ in the sparsity graph of $\Theta$.

We are now ready to present our main result.

**Theorem 1.3** (Main Theorem (Structure learning))**.** *There exists a polynomial-time algorithm which, given a Glauber trajectory from an $(\alpha, d)$-sparse Gaussian graphical model on $n$ vertices, recovers the sparsity pattern (i.e., $\mathrm{supp}(\Theta)$) with probability at least $1 - \delta$, provided the trajectory length satisfies*

$$T = \Omega\left(\frac{d^3 \log(n/\delta)}{\alpha^5}\right).$$

A few remarks are in order. First, the guarantee makes no mixing-time assumption and does not require any additional regularity conditions on $\Theta$ beyond sparsity and the edge-strength condition in Definition 1.1. Second, in the continuous-time dynamics of Definition 1.2, the chain performs $n$ single-site updates per unit time in expectation; thus, observing the chain up to time horizon $T$ corresponds to about $nT$ single-site updates on average. Third (on optimality), information-theoretic lower bounds of $\Omega(\log(d)/\alpha^2)$ were previously known (see Theorem 2 in (Tirukkonda et al., 2025)). Their statement is parameterized by $\beta_{\min} := \min_{(i,j) \in E} |\Theta_{ij}|/\Theta_{ii}$, whereas we work with the symmetric normalization $\min_{(i,j) \in E} |\Theta_{ij}|/\sqrt{\Theta_{ii}\Theta_{jj}}$. Under the condition that $\Theta$ has diagonal 1, these parameters are equal. In Section E (Theorem E.3) using similar techniques, we show an improved information-theoretic lower bound that $\Omega(n \log(n)/\alpha^2)$ single-site updates are necessary for structure learning from a single trajectory (for success probability $1/2$). Since the number of single-site updates and the continuous-time trajectory length are equivalent up to a factor of $n$, this implies a lower bound of $\Omega(\log(n)/\alpha^2)$ on the continuous-time trajectory length.[2] Thus, while our result matches the correct logarithmic dependence, its polynomial dependence on $d$ and $\alpha$ may not be optimal. Closing the remaining gap between the lower bound and the current no-mixing upper bounds is an interesting open problem.

---

[2]See Section G for a formal statement of this equivalence.

Theorem 1.3 follows from the more general parameter-learning theorem below. Our parameter-learning theorem, Theorem 1.4, learns each coordinate up to multiplicative factor $\varepsilon$. The structure-learning theorem then follows by choosing $\varepsilon = O(1)$ and outputting an edge if $|\widehat{\Theta}_{ij}| > \frac{\alpha}{2}$.

**Theorem 1.4** (Main Theorem (Parameter learning))**.** *Let $0 < \alpha, \delta, \varepsilon < 1$ and let $n, d \in \mathbb{N}$. Given a Glauber trajectory evolving according to an $(\alpha, d)$-sparse GGM with precision matrix $\Theta$ and whose length satisfies*

$$T = \Omega\left(\frac{d^3 \log(n/\delta)}{\alpha^5 \varepsilon^5}\right),$$

*there is a polynomial-time algorithm that outputs an estimate $\widehat{\Theta}$ such that*

$$\left|\widehat{\Theta}_{ij} - \Theta_{ij}\right| \leq \varepsilon |\Theta_{ij}| \quad \forall i, j,$$

*with success probability $1 - \delta$.*

We also show the following result when explicit dependence on the mixing time is allowed, without requiring any assumption on the precision matrix beyond the nondegeneracy condition.

**Theorem 1.5** (Structure Learning with Mixing)**.** *There is a polynomial-time algorithm with the following guarantee. For some absolute constant $c > 0$, given a length-$T$ Glauber trajectory from an $(\alpha, d)$-sparse Gaussian graphical model on $n$ vertices, the algorithm recovers the sparsity pattern $\mathrm{supp}(\Theta)$ with probability at least $1 - \delta$, provided the trajectory length satisfies*

$$T = \Omega\left(\frac{\log(n/\delta)}{\alpha^2}\left(t_{\mathrm{mix}}(c\alpha) + \frac{d^2}{\alpha^2}\right)\right).$$

The proof uses the shorter "$iji$" pattern together with mixing gaps between accepted windows. On a clean nearly stationary window, the edge signal appears as the regression coefficient between the $j$-increment and the later $i$-increment, so after conditioning on a large $j$-increment the resulting ratio statistic is a bounded-variance estimate of $\Theta'_{ij}$.

In particular, when the mixing time is guaranteed to be small, for example $t_{\mathrm{mix}}(c\alpha) = O(\log(1/\alpha))$, the bound becomes $T = O\left(\frac{d^2 \log(n/\delta)}{\alpha^4}\right)$. This improves the bound in Theorem 1.3 by a factor of $d/\alpha$. Moreover, relative to the i.i.d. setting, the algorithm effectively requires $\log(n/\delta)/\alpha^2$ nearly independent samples, matching the lower bound for i.i.d. data shown in (Wang et al., 2010).

Similar results with mixing were obtained previously in (Tirukkonda et al., 2025). Compared with that work, Theorem 1.5 gives an improved dependence on $\log(1/\delta)$ and does not require any additional assumptions on the precision matrix. We discuss the extra assumptions in (Tirukkonda et al., 2025) in Section 1.2.

Finally we discuss the practicality of our algorithms.

*Remark* 1.6 (On Practicality). Despite our asymptotic improvements in trajectory length, constants that appear in our analysis preclude the practicality of our algorithm. We believe that these constants are not significantly improvable, and indeed, our experiments suggest that the trajectory lengths required by our algorithms are too large to be practical. We briefly explain the theoretical necessity for these large constants at the end of Section C. Obtaining practical algorithms for this problem is an interesting direction for future work.

To see a full proof of Theorems 1.3 and 1.4 see Section C. For a full proof of Theorem 1.5, see Section D. For a technical overview of the algorithm see Section 2.

## 1.2. Related work

**Gaussian GGMs from Glauber dynamics.** Most closely related to our work is the recent work of Tirukkonda, Rayas, and Dasarathy (Tirukkonda et al., 2025), which gives the first algorithmic guarantees for structure learning in Gaussian graphical models from a single Glauber trajectory, along with information-theoretic lower bounds. In Section H we discuss a technical issue in the proof of one of their lemmas; for now, we summarize their assumptions and guarantees as stated. They impose several regularity conditions beyond a minimum edge-strength assumption. Concretely, in addition to requiring $|\Theta_{ij}|/\Theta_{ii} \geq \beta_{\min}$ for all $(i, j) \in E$, they assume (i) an upper bound $|\Theta_{ij}|/\Theta_{ii} \leq \beta_{\max}$ for all $(i, j) \in E$ (Assumption A1), (ii) bounded marginals $\Sigma_{ii} \leq \sigma_{\max}^2$ and non-degenerate conditionals $1/\Theta_{ii} \geq \sigma_{\min}^2$ (Assumption A2), and (iii) a sample decay condition $d\,\beta_{\max} < 1 - \Omega(1)$ (Assumption A3).[3] Meanwhile, we make no additional assumptions beyond the minimum edge-strength assumption. Most importantly, in the diagonal-1 and $d = O(1)$ setting, the last assumption implies very fast mixing via the Gershgorin circle theorem: $\lambda_{\min}(\Theta) \geq 1 - d\beta_{\max} = \Omega(1)$ and $\lambda_{\max}(\Theta) \leq 1 + d\beta_{\max} = O(1)$ (see Equation (1)). Therefore, under their assumptions one can essentially simulate approximate i.i.d. samples by observing the Glauber trajectory at constant time intervals. However, in our setting, the mixing time may be arbitrarily slow.

**Learning graphical models from Glauber dynamics.** A growing line of work studies structure learning when observations arise from local Markov dynamics rather than i.i.d. sampling. Bresler, Gamarnik, and Shah (Bresler et al., 2017) initiated this direction for discrete graphical models, showing that observing a single-site Glauber trajectory can make structure learning computationally tractable for Ising

models. More recently, Gaitonde and Mossel (Gaitonde & Mossel, 2024) gave a unified analysis for learning Ising models from Glauber trajectories, and Gaitonde, Moitra, and Mossel (Gaitonde et al., 2025a) gave the first efficient algorithms in the observation model that reveals only actual configuration changes rather than all update attempts, with extensions to reversible single-site chains such as Metropolis. In a different direction, Gaitonde, Moitra, and Mossel (Gaitonde et al., 2025b) show that for higher-order Markov random fields, access to Glauber dynamics trajectories can bypass i.i.d. hardness barriers (e.g., noisy parity), yielding efficient learning.

Technically, our algorithm is close in spirit to several methods in this line of work: we examine short windows of the trajectory to probe the interaction between a candidate pair $(i, j)$, and we exploit sparsity to ensure that with sufficiently large probability no neighbor of $i$ or $j$ updates within the window, allowing the local effect of $(i, j)$ to be isolated from confounding updates. That said, structure learning for GGMs differs in important ways from the Ising setting. First, the variables are continuous, so one cannot rely on discrete indicator statistics present in previous work whose empirical averages directly estimate event probabilities. Second, in the Gaussian case a key difficulty is *anti-concentration*: to detect an edge $(i, j)$ one needs the neighbor's influence on the conditional mean to be typically non-negligible. Unlike the Ising setting—where boundedness and discrete concentration can often be leveraged—Gaussian anti-concentration depends on the scale of the conditional variance (equivalently, on $\Theta_{ii}$) and can degrade without additional regularity beyond $(\alpha, d)$. As a result, while the high-level philosophy is shared, the Gaussian case requires new technical ideas.

## 2. Technical Overview

At a high level, our algorithm has three ingredients. First, we estimate the diagonal entries $\Theta_{ii}$ from short windows of the trajectory and use them to normalize the model so that the precision matrix has (approximately) unit diagonal. Second, for each candidate edge $(i, j)$, we look for short update patterns that isolate the influence of $j$ on a later update of $i$. Third, because we cannot directly tell whether hidden neighbor updates occurred inside a window, we treat such windows as contaminated and aggregate many windows using robust median-based estimators.

We begin with diagonal estimation because it already contains the main ideas of the full algorithm: short informative windows, unobserved contamination, and robust aggregation. We then explain how to estimate off-diagonal entries, and finally describe a simpler variant that is available when the chain is allowed to mix between samples.

---

[3]As stated it is written as $d\,\beta_{\max} < 1$; however, the proof of Lemma 4 explicitly uses that this gap is at least a constant.

**Two nearby $i$-updates reveal $\Theta_{ii}$.** Suppose that at some point in the trajectory the $i$th coordinate updates twice, with no update to any neighbor of $i$ in between. Write the corresponding states as

$$X^{(0)} \xrightarrow{i} X^{(1)} \xrightarrow{i} X^{(2)}.$$

Since no neighbor of $i$ changes between the two $i$-updates, both updates use the same conditional mean. Hence

$$X_i^{(1)} - X_i^{(2)} \sim \mathcal{N}\left(0, \frac{2}{\Theta_{ii}}\right).$$

So every such window gives a sample whose variance is exactly $2/\Theta_{ii}$. This turns diagonal estimation into a one-dimensional robust variance-estimation problem.

**The "ii" pattern and hidden contamination.** Exact consecutive "ii" updates are too rare to be useful on the trajectory lengths we target. Instead, we divide the trajectory into short windows of length $T$ and keep every window that contains at least two updates to coordinate $i$. A window is *clean* if, in addition, no neighbor of $i$ updates inside that window. On a clean window, the same calculation as above shows that the difference between the two updated values of coordinate $i$ is distributed as $\mathcal{N}(0, 2/\Theta_{ii})$.

The difficulty is that cleanliness depends on the unknown neighborhood of $i$, so we cannot test it directly. We therefore keep all windows with two $i$-updates and view the non-clean ones as contaminated samples. Because the window is short and the graph is $d$-sparse, the contamination rate is small. Taking the median absolute deviation of the resulting samples gives a robust estimate of $\Theta_{ii}$.

**Normalization.** Using a small initial portion of the trajectory, we estimate every diagonal entry $\Theta_{ii}$. We then rescale coordinate $i$ by $\sqrt{\Theta_{ii}}$ (or, in the actual algorithm, by its estimate), replacing each state $X \in \mathbb{R}^n$ with $X'$ defined by $X_i' = \sqrt{\Theta_{ii}}\, X_i$. Under exact normalization, $X'$ is again a Glauber trajectory, now for a precision matrix with diagonal entries equal to $1$. With estimated diagonals, the normalized trajectory has diagonal entries within $1 \pm \varepsilon$, and the later analysis is stable to this approximation. For the overview, we therefore assume from this point onward that the diagonal is exactly $1$.

**A naive "iji" test for $\Theta_{ij}$.** To estimate an off-diagonal entry $\Theta_{ij}$, the most natural idea is to look for a short window of the form

$$X^{(1)} \xrightarrow{i} X^{(2)} \xrightarrow{j} X^{(3)} \xrightarrow{i} X^{(4)},$$

again with no neighbor of $i$ or $j$ updating inside the window. On such a clean window,

$$X_i^{(2)} = \sum_{k \neq i} -\Theta_{ik} X_k^{(1)} + \varepsilon^{(2)},$$

$$X_i^{(4)} = \sum_{k \neq i} -\Theta_{ik} X_k^{(3)} + \varepsilon^{(4)},$$

with $\varepsilon^{(2)}, \varepsilon^{(4)} \sim \mathcal{N}(0,1)$ i.i.d. The only relevant change between the two conditional means is the value of coordinate $j$, so

$$X_i^{(4)} - X_i^{(2)} = -\Theta_{ij}\big(X_j^{(3)} - X_j^{(1)}\big) + \varepsilon^{(4)} - \varepsilon^{(2)}.$$

This suggests estimating $\Theta_{ij}$ from the ratio

$$\frac{X_i^{(4)} - X_i^{(2)}}{X_j^{(3)} - X_j^{(1)}}.$$

To keep the denominator well behaved, we further condition on $|X_j^{(3)} - X_j^{(1)}| > c$ for a fixed constant $c > 0$.

**Why the "iji" test fails.** The problem is a subtle dependence issue. In the "iji" pattern, the middle $j$-update depends on the value produced by the first $i$-update. As a result, the noise term from the first $i$-update is not independent of the denominator $X_j^{(3)} - X_j^{(1)}$. So the ratio above is not centered in the simple way suggested by the heuristic calculation. This is the main obstacle that forces us away from the naive "iji" statistic.

**The "iiji" pattern breaks the dependence.** To remove this dependence, we prepend one extra $i$-update and instead search for windows of the form

$$X^{(0)} \xrightarrow{i} X^{(1)} \xrightarrow{i} X^{(2)} \xrightarrow{j} X^{(3)} \xrightarrow{i} X^{(4)},$$

where the visible event requires that $i$ updates in the first, second, and fourth quarters, and that $j$ updates in the third quarter, with no off-pattern $i/j$ updates. A window is clean if, in addition, no coordinate in $(N(i) \cup N(j)) \setminus \{i,j\}$ updates in the window. Conditioned on $X^{(0)}$, the extra $i$-update acts as a refresh step: the second $i$-update is independent of the first, so the noise introduced at $X^{(1)}$ is independent of the later state of coordinate $j$. This yields the relation

$$X_i^{(4)} - X_i^{(1)} = -\Theta_{ij}\big(X_j^{(3)} - X_j^{(0)}\big) + \varepsilon^{(4)} - \varepsilon^{(1)},$$

where $\varepsilon^{(1)}, \varepsilon^{(4)} \sim \mathcal{N}(0,1)$ are independent. After conditioning on $\big|X_j^{(3)} - X_j^{(0)}\big| > c$, the ratio statistic

$$\frac{X_i^{(4)} - X_i^{(1)}}{X_j^{(3)} - X_j^{(0)}} = -\Theta_{ij} + \frac{\varepsilon^{(4)} - \varepsilon^{(1)}}{X_j^{(3)} - X_j^{(0)}},$$

is centered at $-\Theta_{ij}$ and has variance bounded by a constant. Thus each clean "iiji" window gives a noisy but informative estimate of $\Theta_{ij}$.

**Robust aggregation.** As in the diagonal-estimation step, we cannot directly verify that a window is clean, because the unknown graph determines which coordinates count as hidden neighbors. We therefore select windows using only the visible "iiji" pattern and treat windows with hidden neighbor updates as contaminated. Moreover, the variance of the ratio statistic can vary from one accepted window to another, and successive windows are not independent because they come from a single trajectory. Nevertheless, the dependence is structured enough that martingale concentration can be used in place of the usual independent-sample Chernoff argument. This shows that the sample median remains a robust estimator of the common location parameter $-\Theta_{ij}$. Estimating every $\Theta_{ij}$ to additive error $O(\alpha)$ is then enough to recover the graph structure. With tighter parameter settings, the same framework also yields multiplicative estimation of the full precision matrix.

**Learning with mixing.** We also give a simpler and more sample-efficient algorithm when the mixing time is allowed to enter the bound. We insert waiting periods of length about $t_{\mathrm{mix}}$ between accepted windows so that each accepted window begins from an almost stationary state. In this regime, the shorter "iji" pattern suffices. On a clean stationary window of the normalized trajectory,

$$X'^{(0)} \xrightarrow{i} X'^{(1)} \xrightarrow{j} X'^{(2)} \xrightarrow{i} X'^{(3)},$$

the pair

$$\left( X_j'^{(2)} - X_j'^{(0)}, \; X_i'^{(3)} - X_i'^{(1)} \right)$$

is centered Gaussian with covariance matrix

$$\begin{pmatrix} 2 & -\Theta'_{ij} \\ -\Theta'_{ij} & 2 \end{pmatrix}.$$

Thus the edge signal appears in the covariance, equivalently in the regression coefficient of the later $i$-increment on the earlier $j$-increment. After passing to the observable trajectory and conditioning on a large $j$-increment, the ratio statistic $-2\widehat{\Delta}_i / \widehat{\Delta}_j$ is a bounded-variance noisy estimate of $\Theta'_{ij}$. A robust median over well-separated epochs then recovers the edge, improving the dependence on $d$ and $\alpha$ when mixing is available.

**Information-theoretic lower bound.** Finally, we prove an information-theoretic lower bound showing that logarithmic trajectory length is unavoidable even without computational constraints. We construct a family of GGMs on $2n$ vertices whose graphs are disjoint unions of edges, with each candidate obtained by deleting one edge from a perfect matching. The resulting trajectories are hard to distinguish from one another. By upper-bounding the pairwise KL divergence and applying Fano's inequality, we obtain a lower bound on the number of Glauber updates required for structure learning.

Our lower bound also differs from (Tirukkonda et al., 2025) in both the hard family and the KL calculation. Instead of working with a broader graph-class construction, we use a simple family obtained from a perfect matching by deleting a single edge. More importantly, we work directly with the exact KL divergence of the Gaussian conditional-update distributions, rather than conditioning on small updates and comparing truncated Gaussians. This yields a $\log n$ improvement in the lower bound. Closing the remaining gap to our current no-mixing upper bounds is an interesting open problem.

# Acknowledgements

The authors thank Jason Gaitonde for helpful discussions and comments on an earlier version of this manuscript, and Jonathan Bloom and Roman Bezrukavnikov for organizing the SPUR program at MIT, where this project began. We thank the anonymous reviewers for their honest feedback, and thank the area chair for their objective and thoughtful judgment.

# Impact Statement

This paper is theoretical and focuses on advancing foundational understanding. As such, it does not present any societal or ethical considerations that require special attention.

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

# A. Preliminaries

## A.1. Continuous-time Glauber dynamics

We consider Glauber dynamics in the continuous-time setting. The initial configuration is arbitrary, and each coordinate $i = 1, \ldots, n$ updates according to an independent Poisson process of rate 1. The following lemma records the relevant update probabilities:

**Lemma A.1** (Lemma 3.1 in (Gaitonde et al., 2025b))**.** *Given an interval $I \subset \mathbb{R}_{\geq 0}$ of length $T$, let $U_I$ denote the set of coordinates that update in $I$. For any $S \subset [n]$ with $|S| = \ell$, we have*

$$\mathbb{P}[S \cap U_I = \varnothing] = \exp(-T\ell),$$
$$and \quad \mathbb{P}[S \subseteq U_I] \geq 1 - \ell \exp(-T).$$

*Proof.* For each Poisson process $\Pi$ of rate 1 and interval $I \subseteq \mathbb{R}_{\geq 0}$, we have $\mathbb{P}[\Pi \cap I = \varnothing] = \exp(-|I|)$, and the claimed bounds follow immediately. $\square$

## A.2. Observing patterns

Throughout the paper, we look for intervals containing a specific "pattern," which we concretely define below:

**Definition A.2** (Pattern exhibition and strictness)**.** A *pattern* of length $k$ is a sequence of indices $P = (i_1, i_2, \ldots, i_k) \in \{1, \ldots, n\}^k$. Consider a pattern $P$ and an interval $I = [t, t+T)$ of a continuous-time single-site update Glauber trajectory. Let $I_j = [t + (j-1)T/k, t + jT/k)$ for $j = 1, \ldots, k$. We say $I$ *exhibits* pattern $P$ if index $i_j$ updates in $I_j$ for each $j = 1, \ldots, k$. We say $I$ is *strict* with respect to $P$ if, for each $j = 1, \ldots, k$, no neighbor of any vertex appearing in $P$, other than $i_j$, updates in $I_j$.

We next bound the probabilities that a given interval exhibits a pattern and is strict with respect to it.

**Lemma A.3.** *Let $I$ be an interval of length $T$, and let $P$ be a pattern containing $\ell$ distinct indices. Then with probability at least $\exp(-T\ell d)$, $I$ is strict with respect to $P$.*

*Proof.* Each index in $P$ has at most $d$ neighbors, so we are forbidding at most $\ell d$ neighbors from updating in each $I_j$. By Lemma A.1, each $I_j$ avoids its respective indices with probability at least $\exp(-T/k \cdot \ell d)$.

The events in the $k$ subintervals are independent, so the probability this holds for all $k$ subintervals is at least $\exp(-T\ell d)$. $\square$

**Lemma A.4.** *Let $I$ be an interval of length $T \leq 1/3$, and let $P$ be a pattern of length $k$. Then with probability at least $T^k/(2k^k)$, $I$ exhibits $P$.*

*Proof.* By Lemma A.1,

$$\mathbb{P}[I \text{ exhibits } P] \geq (1 - \exp(-T/k))^k$$
$$\geq \frac{T^k}{k^k} \exp(-T) \geq \frac{T^k}{2k^k},$$

where we use the estimates $1 - \exp(-x) \geq x \exp(-x)$ and $T \leq 1/3$. $\square$

**Lemma A.5.** *For any interval $I$ and pattern $P$, the event that $I$ exhibits $P$ is independent of the event that $I$ is strict with respect to $P$.*

*Proof.* In each subinterval $I_j$, exhibition depends only on the clock of $i_j$, while strictness depends only on the clocks of the remaining coordinates. Thus the two events are independent in each $I_j$, and the claim follows by independence across the disjoint subintervals. $\square$

## A.3. Corruption and robust estimation

We use several robust estimators in this paper. We work with the following corruption model.

**Definition A.6** ($\eta$-corruption)**.** Let $\mathcal{D}$ be a distribution, and let $X_1, \ldots, X_n \sim \mathcal{D}$ be i.i.d. samples. We say that the observed samples $\widetilde{X}_1, \ldots, \widetilde{X}_n$ are $\eta$-*corrupted* if there exist i.i.d. Bernoulli($\eta$) random variables $\xi_1, \ldots, \xi_n$ such that for each $i = 1, \ldots, n$,

$$\widetilde{X}_i = \begin{cases} X_i, & \text{if } \xi_i = 0 \\ A_i, & \text{if } \xi_i = 1, \end{cases}$$

where each $A_i$ is an arbitrary value. Whenever $\widetilde{X}_i = A_i$, the corrupted value may be chosen adversarially, may depend on the entire trajectory up to the corrupted sample $\widetilde{X}_i$, and need not be drawn from any distribution.[4]

Note that our corruption model differs from the Huber contamination model, as corrupted entries may be selected adversarially. It also differs from the strong contamination model in that each sample is independently corrupted with probability $\eta$. Further, as Glauber trajectories are sampled sequentially, corrupted samples in our corruption model may only depend on prior samples.

Under this corruption model, we use the following robust estimator for variance.

**Lemma A.7** (Robust variance estimation)**.** *Let $0 < \eta \leq 1/10$ and $0 < \delta < 1$. There is a linear-time algorithm that takes $n$ samples from $\mathcal{N}(0, \sigma^2)$, an $\eta$-fraction of which are corrupted (as in Definition A.6), and outputs $\widehat{\sigma}$ such that $|\widehat{\sigma} - \sigma| < 5\eta\sigma$ with probability $1 - \delta$, provided $n \geq \frac{2\log(4/\delta)}{\eta^2}$.*

---

[4]This corruption model is closely related to what some recent work calls *malicious noise* or *strong malicious noise*; see, e.g., (Blanc et al., 2026).

We also use the following mean estimator, which allows for another source of adversarial control in the choice of the variances.

**Lemma A.8** (Robust mean estimation). *Let $0 < \eta \leq 1/10$ and $0 < \delta < 1$. Let $\Phi$ denote the cdf of a standard Gaussian. We are given a filtration $(\mathcal{F}_\ell)_{\ell=0}^n$ and samples $x^{(1)}, \ldots, x^{(n)}$. Suppose there exist indicators $\xi^{(\ell)} \in \{0, 1\}$ such that*

$$\mathbb{E}[\xi^{(\ell)} \mid \mathcal{F}_{\ell-1}] \leq \eta \qquad \text{for all } \ell.$$

*Assume moreover that whenever $\xi^{(\ell)} = 0$, the clean sample is centered at the same value $\mu$ and has Gaussian tails dominated by unit variance in the sense that, for every $t \geq 0$,*

$$\mathbb{P}\Big[x^{(\ell)} \geq \mu + t \,\Big|\, \mathcal{F}_{\ell-1}, \xi^{(\ell)} = 0\Big] \leq 1 - \Phi(t),$$

$$\mathbb{P}\Big[x^{(\ell)} \leq \mu - t \,\Big|\, \mathcal{F}_{\ell-1}, \xi^{(\ell)} = 0\Big] \leq 1 - \Phi(t).$$

*Then the sample median $\widehat{\mu}$ satisfies $|\widehat{\mu} - \mu| < 5\eta$ with probability $1 - \delta$, provided $n \geq \frac{2\log(2/\delta)}{\eta^2}$.*

Note that Lemma A.7 remains valid even when the corrupted samples are chosen adversarially with knowledge of the full trajectory of iterates, whereas Lemma A.8 does not. In particular, Lemma A.8 applies whenever, conditional on $\mathcal{F}_{\ell-1}$ and on the sample being uncorrupted, the clean law is a Gaussian scale mixture $\mu + \sigma\varepsilon$ with $|\sigma| < 1$: averaging over $\sigma$ preserves the displayed tail bounds.

The proofs of these estimators are provided in Section F.

## A.4. Concentration inequalities

We use the following concentration inequalities in the proofs of Lemmas A.7 and A.8.

**Fact A.9** (Dvoretzky–Kiefer–Wolfowitz). *Let $Y_1, \ldots, Y_n$ be i.i.d. real-valued random variables with cdf $F$, and let*

$$F_n(t) := \frac{1}{n}\sum_{i=1}^n \mathbf{1}\{Y_i \leq t\},$$

*be the empirical cdf. Then for every $\varepsilon > 0$,*

$$\mathbb{P}\Big[\sup_{t \in \mathbb{R}}\big|F_n(t) - F(t)\big| > \varepsilon\Big] \leq 2e^{-2n\varepsilon^2}.$$

**Fact A.10** (Azuma–Hoeffding). *Let $(M_k, \mathcal{F}_k)_{k=0}^n$ be a martingale, and suppose that for each $k = 1, \ldots, n$,*

$$M_k - M_{k-1} \in [a_k, b_k] \qquad \text{almost surely.}$$

*Then for every $t > 0$,*

$$\mathbb{P}[M_n - M_0 \geq t] \leq \exp\Big(-\frac{2t^2}{\sum_{k=1}^n (b_k - a_k)^2}\Big).$$

*In particular, if each increment lies in an interval of length at most 1, then*

$$\mathbb{P}[M_n - M_0 \geq t] \leq e^{-2t^2/n}.$$

## A.5. Mixing and coupling

**Definition A.11** (Mixing and mixed chains). A position $X_i$ of the Markov chain $X_0, X_1, \ldots$ is $\varepsilon$-*mixed* if $X_i$ has TV-distance at most $\varepsilon$ from its stationary distribution $\pi$. We define the *mixing time* by

$$t_{\mathrm{mix}}(\varepsilon) = \min\Big\{t \geq 0$$
$$: \max_x d_{\mathrm{TV}}(\mathcal{L}(X_t \mid X_0 = x), \pi) \leq \varepsilon\Big\},$$

where the maximum is over all starting states $x$.

The stationary distribution of a Glauber trajectory with covariance matrix $\Sigma$ is $\mathcal{N}(0, \Sigma)$. We use the following slightly stronger invariance statement.

**Lemma A.12.** *Fix a coordinate $i$. If $x \sim \mathcal{N}(0, \Sigma)$, then after a single Glauber update to the $i$th coordinate, the resulting position $x'$ still satisfies $x' \sim \mathcal{N}(0, \Sigma)$.*

*Proof.* Without loss of generality, we update the first coordinate. Write precision and covariance in block matrix form as

$$\Sigma = \begin{pmatrix} \Sigma_{11} & \Sigma_{1,-1} \\ \Sigma_{-1,1} & \Sigma_{-1,-1} \end{pmatrix},$$
$$\Theta = \begin{pmatrix} \Theta_{11} & \Theta_{1,-1} \\ \Theta_{-1,1} & \Theta_{-1,-1} \end{pmatrix}.$$

The update rule is

$$x'_{-1} = x_{-1}, \qquad x'_1 = -\frac{\Theta_{1,-1}}{\Theta_{11}}x_{-1} + \mathcal{N}\Big(0, \frac{1}{\Theta_{11}}\Big).$$

Then $(x'_1, x'_{-1})$ is still Gaussian with mean 0. Moreover,

$$\mathrm{Cov}(x'_1, x'_{-1}) = -\frac{\Theta_{1,-1}}{\Theta_{11}}\Sigma_{-1,-1} = \Sigma_{1,-1},$$
$$\mathrm{Var}(x'_1) = \frac{1}{\Theta_{11}} + \frac{\Theta_{1,-1}}{\Theta_{11}}\Sigma_{-1,-1}\frac{\Theta_{-1,1}}{\Theta_{11}} = \Sigma_{11},$$

by Schur's complement. Since $x'_{-1} = x_{-1}$, this proves $x' \sim \mathcal{N}(0, \Sigma)$. □

From here, we may use the coupling lemma to derive a similar statement about mixing.

**Fact A.13** (Coupling lemma). *Let $\mu$ and $\eta$ be distributions over $\mathbb{R}^n$. For any coupling $\omega$ of $\mu$ and $\eta$, if $(X, Y) \sim \omega$, then $\mathbb{P}[X \neq Y] \geq d_{\mathrm{TV}}(\mu, \eta)$. Moreover, equality is attained for some coupling.*

**Corollary A.14.** *Let $\mu$ and $\eta$ be distributions. Suppose $f(X) \sim \eta$ whenever $X \sim \mu$. Then, if $Y$ is sampled from a distribution with TV-distance $\varepsilon$ from $\mu$, then $f(Y)$ follows a distribution with TV-distance at most $\varepsilon$ from $\eta$.*

*Proof.* Let $\nu$ denote the distribution of $Y$. By Fact A.13, there exists a coupling of $X \sim \mu$ and $Y \sim \nu$ such that $\mathbb{P}[X \neq Y] = d_{\text{TV}}(\mu, \nu) \leq \varepsilon$. Under this coupling, $f(X) \sim \eta$ and $\mathbb{P}[f(X) \neq f(Y)] \leq \varepsilon$. Another application of Fact A.13 gives the claim. $\square$

**Corollary A.15.** *Fix a coordinate $i$. If a Glauber trajectory is $\varepsilon$-mixed (as in Definition A.11), and undergoes a Glauber update to the $i$th coordinate, it remains $\varepsilon$-mixed.*

*Proof.* Combine Lemma A.12 and Corollary A.14. $\square$

# B. Reduction to Normalized Gaussian Graphical Models

In this section we design an algorithm that reduces the problem of learning an arbitrary sparse Gaussian graphical model from a Glauber trajectory to the case where the diagonal entries of the precision matrix are 1.

To this end, we first estimate the diagonal entries up to a multiplicative factor $(1 \pm O(\varepsilon))$. We then scale the Glauber trajectory $X^{(1)}, X^{(2)}, \ldots$ by $X_i'^{(k)} := \sqrt{\Theta_{ii}} \cdot X_i^{(k)}$ for all $i, k$, which we show is itself a Glauber trajectory. Using our estimates for $\Theta_{ii}$, we then have a coordinate-wise $(1 \pm O(\varepsilon))$-approximation of this scaled trajectory, which also happens to be a Glauber trajectory itself.

Our main result for this section is as follows:

**Theorem B.1** (Normalization). *Let $0 < \alpha, \delta < 1$ and $0 < \varepsilon \leq 1/10$, and let $n, d \in \mathbb{N}$. Given a Glauber trajectory evolving according to an $(\alpha, d)$-sparse GGM with precision matrix $\Theta$ and having length*

$$T = \frac{4000d \log(8n/\delta)}{\varepsilon^3} + T_{\text{rest}},$$

*where $T_{\text{rest}} \geq 0$, there is a polynomial-time algorithm that outputs an estimate $D$ for $\text{diag}(\Theta)$ such that*

$$\left| \frac{1}{\sqrt{D_i}} - \frac{1}{\sqrt{\Theta_{ii}}} \right| \leq \frac{\varepsilon}{\sqrt{\Theta_{ii}}} \quad \forall i,$$

*and a trajectory of length $T_{\text{rest}}$ evolving according to a GGM with precision matrix $\widehat{\Theta} = D^{-1/2} \Theta D^{-1/2}$ with probability $1 - \delta$.*

We proceed in three steps. Section B.1 identifies the "$ii$" statistic underlying diagonal estimation. Section B.2 uses this statistic together with robust variance estimation to prove Lemma B.2. Section B.3 shows that the rescaled trajectory is again a Glauber trajectory and then completes the proof of Theorem B.1.

**Lemma B.2.** *For fixed $i$, given a Glauber trajectory whose length satisfies*

$$T \geq \frac{4000d \log(8/\delta)}{\varepsilon^3},$$

*we may retrieve in polynomial time an estimate $D_i$ for $\Theta_{ii}$ with*

$$\left| \frac{1}{\sqrt{D_i}} - \frac{1}{\sqrt{\Theta_{ii}}} \right| \leq \frac{\varepsilon}{\sqrt{\Theta_{ii}}}$$

*with probability $1 - \delta$.*

## B.1. Properties of the statistic

Fix $i$. Let $T \leq 1/3$, and let $I_t$ denote the time interval $[t, t+T)$. Let $\mathcal{A}^{(t)}$ denote the event that $I_t$ is strict with respect to $(i, i)$, and $\mathcal{B}^{(t)}$ the event $I_t$ exhibits $(i, i)$.

If $\mathcal{A}^{(t)}$ and $\mathcal{B}^{(t)}$ both hold, let $Y^{(0)}, Y^{(1)}, Y^{(2)}$ denote the position at times $t, t + T/2, t + T$ respectively.

**Lemma B.3.** *We have $Y_i^{(1)} - Y_i^{(2)} \mid \mathcal{A}^{(t)}, \mathcal{B}^{(t)} \sim \mathcal{N}\left(0, \frac{2}{\Theta_{ii}}\right)$.*

*Proof.* By Definition 1.2, we have

$$Y_i^{(1)} = -\sum_{j \neq i} -\frac{\Theta_{ij}}{\Theta_{ii}} Y_j^{(0)} + \varepsilon_1,$$

$$\text{and} \quad Y_i^{(2)} = -\sum_{j \neq i} -\frac{\Theta_{ij}}{\Theta_{ii}} Y_j^{(1)} + \varepsilon_2,$$

where $\varepsilon_1, \varepsilon_2 \sim \mathcal{N}(0, \frac{1}{\Theta_{ii}})$ are i.i.d.

If $\mathcal{A}^{(t)}$ holds, $Y_j^{(1)} = Y_j^{(2)}$ for all $j \neq i$, so

$$Y_i^{(1)} - Y_i^{(2)} \mid Y^{(0)}, \mathcal{A}^{(t)}, \mathcal{B}^{(t)}$$
$$= \varepsilon_1 - \varepsilon_2 \sim \mathcal{N}\left(0, \frac{2}{\Theta_{ii}}\right),$$

and the dependence on $Y^{(0)}$ may be dropped. $\square$

## B.2. Retrieving the estimator

We divide our sample into $M$ intervals of length $T$, discarding intervals that do not satisfy $\mathcal{B}^{(t)}$, sampling $Y_i^{(1)} - Y_i^{(2)}$ from the rest, and treating intervals that fail to satisfy $\mathcal{A}^{(t)}$ as corruption in order to predict $1/\Theta_{ii}$ with robust one-dimensional variance estimation.

By Lemmas A.3 and A.5, the corruption rate among the retained intervals is bounded by

$$q_i := \mathbb{P}[\neg \mathcal{A}^{(t)} \mid \mathcal{B}^{(t)}] \leq 1 - e^{-dT}.$$

**Lemma B.4.** *In a Glauber dynamic of length $MT$, in at least $\frac{1}{16} MT^2$ intervals $\mathcal{B}^{(t)}$ holds with probability at least $1 - \exp(-\frac{1}{64} MT^2)$.*

*Proof.* By Lemma A.4, each interval satisfies $\mathcal{B}^{(t)}$ with probability at least $T^2/8$. These events are independent

across disjoint intervals. A Chernoff lower-tail bound therefore gives

$$\mathbb{P}\left[\sum_{r=1}^{M}\mathbf{1}\{\mathcal{B}^{(r)}\} \leq \frac{1}{2}\frac{MT^2}{8}\right] \leq \exp\left(-\frac{1}{8}\frac{MT^2}{8}\right),$$

which is exactly the claimed bound. $\qquad\square$

*Proof of Lemma B.2.* Choose

$$T = \frac{\varepsilon}{5d} \qquad \text{and} \qquad MT^2 = \frac{800\log(8/\delta)}{\varepsilon^2}.$$

Then the actual corruption rate satisfies

$$q_i \leq 1 - e^{-dT} \leq 1 - e^{-\varepsilon/5} \leq \frac{\varepsilon}{5}.$$

Since

$$\exp\left(-\frac{MT^2}{64}\right) = \exp\left(-\frac{25\log(8/\delta)}{2\varepsilon^2}\right) \leq \frac{\delta}{8},$$

by Lemma B.4, with probability $1 - \delta/2$ we see

$$\frac{MT^2}{16} = \frac{50\log(8/\delta)}{\varepsilon^2} = \frac{2\log(8/\delta)}{(\varepsilon/5)^2}.$$

Then, by Lemma A.7 with corruption parameter $q_i \leq \varepsilon/5$ and failure probability $\delta/2$, we can estimate $\sqrt{2/\Theta_{ii}}$ within additive error

$$5 \cdot \frac{\varepsilon}{5}\sqrt{\frac{2}{\Theta_{ii}}} = \varepsilon\sqrt{\frac{2}{\Theta_{ii}}},$$

and hence within a factor $1 \pm \varepsilon$, with success probability $1 - \delta/2$. Taking a union bound gives the desired success rate.

Finally, the required length of the trajectory is

$$MT = \frac{4000d\log(8/\delta)}{\varepsilon^3}. \qquad\square$$

Finally, we derive estimates for all $n$ diagonal entries $\Theta_{ii}$.

**Corollary B.5.** *Given a Glauber dynamic of length*

$$T = \frac{4000d\log(8n/\delta)}{\varepsilon^3},$$

*we may retrieve in polynomial time estimates $D_i$ for each $i$ such that, with probability at least $1 - \delta$,*

$$\left|\frac{1}{\sqrt{D_i}} - \frac{1}{\sqrt{\Theta_{ii}}}\right| \leq \frac{\varepsilon}{\sqrt{\Theta_{ii}}}$$

*for all $i = 1, \ldots, n$.*

*Proof.* Apply Lemma B.2 with error $\delta/n$. Then, the probability of any error among the $n$ estimates is $\delta$ by union bound. $\qquad\square$

---

**Algorithm 1** $D = \text{EstimateDiagonal}(n, d, \alpha, \varepsilon, \delta)$

---

1: Set $T = \frac{\varepsilon}{5d}$ and $M = \frac{20000d^2\log(8n/\delta)}{\varepsilon^4}$.
2: Observe Glauber trajectory of length $MT$, split into $M$ intervals $I_1, \ldots, I_M$ of length $T$.
3: **for** $i = 1, \ldots, n$ **do**
4:     Let $S = \{1 \leq t \leq M : \mathcal{B}^{(t)}\}$, where $\mathcal{B}^{(t)}$ is defined in Section B.1.
5:     **if** $|S| < \frac{1}{16}MT^2$ **then**
6:         Return $\perp$.
7:     **else**
8:         By Lemma A.7, estimate $\widehat{\sigma}^2$ from $\{Y_i^{(1)} - Y_i^{(2)} : t \in S\}$, where $Y_i^{(1)}$ and $Y_i^{(2)}$ are defined in Section B.1.
9:         Set $D_i = 2/\widehat{\sigma}^2$.
10:     **end if**
11: **end for**
12: Return $D$.

---

### B.3. Normalizing the Gaussian

Let $X^{(1)}, X^{(2)}, \ldots$ denote the updates from the Glauber dynamic. As earlier, define $X'^{(1)}, X'^{(2)}, \ldots$ so that $X_i'^{(k)} := \sqrt{\Theta_{ii}} \cdot X_i^{(k)}$ for all $i, k$.

**Lemma B.6.** *The above $X'^{(1)}, X'^{(2)}, \ldots$ is a Glauber trajectory with precision matrix $\Theta' = \Theta_{\text{diag}}^{-1/2}\Theta\Theta_{\text{diag}}^{-1/2}$.*

*Proof.* Note that $\Theta'_{ij} = \frac{\Theta_{ij}}{\sqrt{\Theta_{ii}\Theta_{jj}}}$.

We may directly check that Definition 1.2 is preserved. We have

$$x_i' \mapsto \sqrt{\Theta_{ii}}\left(-\sum_{j=1}^{n}\frac{\Theta_{ij}}{\Theta_{ii}}\frac{x_j'}{\sqrt{\Theta_{jj}}} + \mathcal{N}\left(0, \frac{1}{\Theta_{ii}}\right)\right)$$

$$= -\sum_{j=1}^{n}\frac{\Theta_{ij}}{\sqrt{\Theta_{ii}\Theta_{jj}}}x_j' + \mathcal{N}(0, 1). \qquad\square$$

To prove our main theorem, we consider the trajectory $\widehat{X}^{(1)}, \widehat{X}^{(2)}, \ldots$ defined by $\widehat{X}_i^{(k)} = \sqrt{D_i} \cdot X_i^{(k)}$ for all $i, k$.

### Putting the normalization together.

*Proof of Theorem B.1.* It suffices to show $\widehat{X}$ is a Glauber dynamic with precision matrix $\widehat{\Theta} = D^{-1/2}\Theta D^{-1/2}$, i.e. $\widehat{\Theta}_{ij} = \frac{\Theta_{ij}}{\sqrt{D_iD_j}}$. Again, we check Definition 1.2 is preserved.

We have

$$\widehat{x}_i \mapsto \sqrt{D_i} \left( -\sum_{j=1}^{n} \frac{\Theta_{ij}}{\Theta_{ii}} \frac{\widehat{x}_j}{\sqrt{D_j}} + \mathcal{N}\left(0, \frac{1}{\Theta_{ii}}\right) \right)$$

$$= -\sum_{j=1}^{n} \frac{\widehat{\Theta}_{ij}}{\widehat{\Theta}_{ii}} \widehat{x}_j + \mathcal{N}\left(0, \frac{1}{\widehat{\Theta}_{ii}}\right).$$

Finally, for independence reasons, we use only the first

$$T_{\text{diag}} := \frac{4000d \log(8n/\delta)}{\varepsilon^3}$$

updates of the trajectory to estimate $D$, discard that initial segment, and keep the remaining trajectory of length $T_{\text{rest}}$ to produce the trajectory with the desired properties. $\quad\square$

In particular, note that by construction, $\widehat{\Theta}$ is coordinate-wise a $(1 \pm O(\varepsilon))$-approximation for $\Theta'$ (defined in Lemma B.6).

Concretely,

**Corollary B.7.** *There exist* $c_1, \ldots, c_n \in 1 \pm \varepsilon$ *so that for each* $i$ *and* $k$, *we have* $\widehat{X}_i^{(k)} = \frac{1}{c_i} X_i'^{(k)}$; *moreover* $\widehat{\Theta}_{ij} = c_i c_j \Theta_{ij}'$ *for each* $i$ *and* $j$.

*Proof.* Indeed,

$$\frac{X_i'^{(k)}}{\widehat{X}_i^{(k)}} = \sqrt{\frac{\Theta_{ii}}{D_i}} =: c_i \in 1 \pm \varepsilon,$$

and further

$$\frac{\widehat{\Theta}_{ij}}{\Theta_{ij}'} = \frac{\sqrt{\Theta_{ii}\Theta_{jj}}}{\sqrt{D_i D_j}} = c_i c_j. \quad\square$$

Thus, the trajectory $\widehat{X}$ we constructed has two properties:

- it is itself a Glauber trajectory with diagonal entries $1 \pm O(\varepsilon)$ (and precision matrix $\widehat{\Theta}$);

- it is coordinate-wise a $(1 \pm \varepsilon)$-approximation for a Glauber trajectory with diagonal entries $1$ (and precision matrix $\Theta'$).

## C. Main Algorithm

We now move to structure-learning. Our main result is as follows:

**Theorem C.1** (Theorem 1.3)**.** *Let* $0 < \alpha, \delta < 1$ *and* $n, d \in \mathbb{N}$. *Given a Glauber trajectory evolving according to an* $(\alpha, d)$-*sparse GGM with precision matrix* $\Theta$ *and whose length satisfies*

$$T \geq \frac{256 \cdot 257^5 \, d^3 \log(4n/\delta)}{\alpha^5},$$

*there is a polynomial-time algorithm that correctly outputs whether* $i \sim j$ *for each* $i$ *and* $j$ *with probability* $1 - \delta$.

We also derive the following parameter-learning guarantees:

**Corollary C.2** (Theorem 1.4)**.** *Let* $0 < \alpha, \delta, \varepsilon < 1$, *and let* $n, d \in \mathbb{N}$. *Given a Glauber trajectory evolving according to an* $(\alpha, d)$-*sparse GGM with precision matrix* $\Theta$ *and whose length satisfies*

$$T \geq \frac{256 \cdot 257^5 \, d^3 \log(4n/\delta)}{\alpha^5 \varepsilon^5},$$

*there is a polynomial-time algorithm that outputs an estimate* $\widehat{\Theta}$ *for* $\Theta$ *such that* $\left| \widehat{\Theta}_{ij} - \Theta_{ij} \right| \leq \varepsilon \left| \Theta_{ij} \right|$ *for each* $i$ *and* $j$.

To this end, we fix $i$ and $j$, and evaluate the existence of each edge $i \sim j$ individually.

**Lemma C.3.** *For fixed* $i$ *and* $j$, *given a Glauber trajectory whose length satisfies*

$$T \geq \frac{128 \cdot 257^5 \, d^3 \log(16/\delta)}{\alpha^5},$$

*we may determine whether* $i \sim j$ *in polynomial time with probability* $1 - \delta$.

Section C.1 analyzes the statistic used to test whether $i \sim j$: first in the ideal normalized trajectory, then for the observable approximate trajectory, and finally after conditioning on a large denominator. Section C.2 uses these ingredients to build the estimator and complete the proofs of Lemma C.3, Theorem C.1, and Corollary C.2.

A naive analysis based on the shorter "$iji$" pattern fails because the middle $j$-update depends on the earlier $i$-update, creating a dependence in the resulting ratio statistic. For this reason we instead study the "$iiji$" pattern, whose extra $i$-update breaks this dependence and leads to the statistic analyzed below. The observations remain dependent and can be corrupted by hidden neighbor updates, so after conditioning on $|\Delta_j| > c$ we estimate the resulting location parameter using Lemma A.8.

### C.1. Properties of the statistic

Fix $i$ and $j$. Let $T \leq 1/3$, let $I_t$ denote the time interval $[t, t+T)$, and let $P_{iiji} := (i, i, j, i)$. Let $\mathcal{D}^{(t)}$ denote the observable event that $I_t$ exhibits $P_{iiji}$, and let $\mathcal{C}^{(t)}$ denote the event that $I_t$ is strict with respect to $P_{iiji}$. By Lemma A.5, $\mathcal{D}^{(t)}$ and $\mathcal{C}^{(t)}$ are independent.

Normalize the Glauber trajectory by Theorem B.1, and let $\widehat{X}$ consist of $(1 \pm 1/10)$ coordinate-wise approximations of $X'$. We first analyze the normalized Glauber dynamic $X'$.

**The ideal normalized statistic.** Conditioned on $\mathcal{C}^{(t)}$ and $\mathcal{D}^{(t)}$, let $Y'^{(k)}$ denote the value of $X'$ at time $t + Tk/4$, for

$k = 0, 1, 2, 3, 4$. Denote

$$\Delta_j'^{(t)} := Y_j'^{(3)} - Y_j'^{(0)} \quad \text{and} \quad \Delta_i'^{(t)} := Y_i'^{(4)} - Y_i'^{(1)}.$$

**Lemma C.4.** *We have* $\Delta_i'^{(t)} \mid \mathcal{C}^{(t)}, \mathcal{D}^{(t)}, Y'^{(0)}, Y'^{(3)} \sim -\Theta_{ij}' \Delta_j'^{(t)} + \mathcal{N}(0, 2)$.

*Proof.* Throughout this proof we condition on $\mathcal{C}^{(t)}$ and $\mathcal{D}^{(t)}$. Let $\varepsilon_1, \varepsilon_2, \varepsilon_3, \varepsilon_4 \sim \mathcal{N}(0, 1)$ denote the fresh noises coming from the last designated updates of $i$ in the first, second, and fourth quarters, and of $j$ in the third quarter, respectively. By the definition of $\mathcal{D}^{(t)}$ and $\mathcal{C}^{(t)}$, the interval exhibits $P_{iiji}$ and is strict with respect to it.

First,

$$Y_i'^{(1)} = -\sum_{k \neq i} \Theta_{ik}' Y_k'^{(0)} + \varepsilon_1,$$

because no neighbor of $i$ changes during the first quarter. Likewise,

$$Y_i'^{(2)} = -\sum_{k \neq i} \Theta_{ik}' Y_k'^{(0)} + \varepsilon_2,$$

because the second quarter again contains an $i$-update and no neighbor of $i$ updates.

Before the designated $j$-update in the third quarter, the only neighbor of $j$ that may have changed is $i$, so

$$Y_j'^{(3)} = -\sum_{k \neq j} \Theta_{jk}' Y_k'^{(2)} + \varepsilon_3.$$

Thus $Y'^{(3)}$ depends on $\varepsilon_2$ and $\varepsilon_3$, but not on $\varepsilon_1$.

Finally, before the designated $i$-update in the fourth quarter, the only neighbor of $i$ that may have changed is $j$, so

$$Y_i'^{(4)} = -\sum_{k \neq i,j} \Theta_{ik}' Y_k'^{(0)} - \Theta_{ij}' Y_j'^{(3)} + \varepsilon_4.$$

Subtracting the expression for $Y_i'^{(1)}$ gives

$$Y_i'^{(4)} - Y_i'^{(1)} = -\Theta_{ij}' \big(Y_j'^{(3)} - Y_j'^{(0)}\big) + \varepsilon_4 - \varepsilon_1.$$

Then $\varepsilon_4 - \varepsilon_1 \sim \mathcal{N}(0, 2)$ is independent of $Y'^{(0)}$ and $Y'^{(3)}$. This proves the claim. $\square$

**Passing to the observable statistic.** We now turn to the observable data from $\widehat{X}$. Let $\widehat{Y}^{(k)}$ denote the value of $\widehat{X}$ at time $t + Tk/4$, for $k = 0, 1, 2, 3, 4$. Denote

$$\widehat{\Delta}_j^{(t)} = \widehat{Y}_j^{(3)} - \widehat{Y}_j^{(0)} \quad \text{and} \quad \widehat{\Delta}_i^{(t)} = \widehat{Y}_i^{(4)} - \widehat{Y}_i^{(1)}.$$

**Lemma C.5.** *For constants* $c_i, c_j \in 1 \pm 1/10$ *as in Corollary* B.7*, we have for all $t$ with $\mathcal{C}^{(t)}$ and $\mathcal{D}^{(t)}$ that*

$$\widehat{\Delta}_i^{(t)} \mid \mathcal{C}^{(t)}, \mathcal{D}^{(t)}, \widehat{Y}^{(0)}, \widehat{Y}^{(3)}$$

$$\sim -\frac{c_j}{c_i} \Theta_{ij}' \widehat{\Delta}_j^{(t)} + \mathcal{N}\left(0, \frac{2}{c_i^2}\right).$$

*Proof.* By Corollary B.7, we have $\widehat{\Delta}_i^{(t)} = \frac{1}{c_i} \Delta_i'^{(t)}$ and $\widehat{\Delta}_j^{(t)} = \frac{1}{c_j} \Delta_j'^{(t)}$, from which the claim follows. $\square$

**Conditioning on a large denominator.** Let $\mathcal{E}^{(t)}$ denote the condition that $|\widehat{\Delta}_j^{(t)}| \geq 0.6$. We will restrict our attention to samples where $\mathcal{E}^{(t)}$ holds.

**Lemma C.6.** *We have* $\mathbb{P}[\mathcal{E}^{(t)} \mid \mathcal{D}^{(t)}, \widehat{Y}^{(0)}] > 1/4$.

*Proof.* As $c_j < 1.1$, we have that $|\widehat{\Delta}_j^{(t)}| \geq 0.6$ is implied by $|\Delta_j'^{(t)}| \geq 0.66$, so it suffices to show $|\Delta_j'^{(t)}| \geq 0.66$ with probability more than $1/4$.

Let $Z$ be the value of $X'$ right before the last $j$-update in the third quarter. Then, by Definition 1.2, $Y_j'^{(3)}$ given $Z$ is a Gaussian with variance 1, i.e. $\Delta_j'^{(t)} \mid Z \sim \mathcal{N}(m, 1)$ for some $m$ dependent on $Z$.

Therefore, for any fixed $Z$, we have

$$\begin{aligned} \mathbb{P}[|\Delta_j'^{(t)}| > 0.66 \mid Z] &= \mathbb{P}_{x \sim \mathcal{N}(m,1)}[|x| \geq 0.66 \mid Z] \\ &\geq \mathbb{P}_{x \sim \mathcal{N}(|m|,1)}[x \geq 0.66 \mid Z] \\ &\geq \mathbb{P}_{x \sim \mathcal{N}(0,1)}[x \geq 0.66] > 1/4, \end{aligned}$$

implying that

$$\mathbb{P}[|\Delta_j'^{(t)}| > 0.66 \mid \mathcal{D}^{(t)}, \widehat{Y}^{(0)}] > \frac{1}{4},$$

regardless of the realized value of $Z$. $\square$

**Lemma C.7.** *For some $\sigma < 2.62$, we have*

$$-\frac{\widehat{\Delta}_i^{(t)}}{\widehat{\Delta}_j^{(t)}} \;\Bigg|\; \widehat{Y}^{(0)}, \widehat{Y}^{(3)}, \mathcal{C}^{(t)}, \mathcal{D}^{(t)}, \mathcal{E}^{(t)} \sim \frac{c_j}{c_i} \Theta_{ij}' + \mathcal{N}(0, \sigma^2).$$

*Proof.* Since $\mathcal{E}^{(t)}$ is determined by $\widehat{Y}^{(0)}$ and $\widehat{Y}^{(3)}$, Lemma C.5 gives

$$\frac{\widehat{\Delta}_i^{(t)}}{\widehat{\Delta}_j^{(t)}} \;\Bigg|\; \widehat{Y}^{(0)}, \widehat{Y}^{(3)}, \mathcal{C}^{(t)}, \mathcal{D}^{(t)}, \mathcal{E}^{(t)}$$

$$\sim -\frac{c_j}{c_i} \Theta_{ij}' + \mathcal{N}\left(0, \frac{2}{c_i^2 \left(\widehat{\Delta}_j^{(t)}\right)^2}\right),$$

Since $c_i > 0.9$ and $|\widehat{\Delta}_j^{(t)}| > 0.6$, the upper bound follows from $\frac{\sqrt{2}}{c_i |\widehat{\Delta}_j^{(t)}|} < 2.62$. $\square$

### C.2. Retrieving the estimator

We divide our Glauber trajectory into $M$ intervals of length $T$, for some $M$ and $T$ we will decide later (so the trajectory has length $MT$). We discard intervals that do not satisfy

---

**Algorithm 2** $\widetilde{\Theta}$ = LearnGGM$(n, d, \alpha, \varepsilon, \delta)$

---

1: Set $\phi = \alpha/257$, $T = \phi/d$, and $M = 256d^4 \log(4n/\delta)/\phi^6$.
2: Let $\text{diag}(\widetilde{\Theta}) = \text{EstimateDiagonal}(n, d, \alpha, 1/10, \delta/2)$ from Algorithm 1.
3: Use the remaining trajectory.
4: Let $E = \varnothing$.
5: Observe Glauber trajectory of length $MT$, split into $M$ intervals $I_1, \ldots, I_M$ of length $T$.
6: **for** $i = 1, \ldots, n$ **do**
7:    **for** $j = i + 1, \ldots, n$ **do**
8:       Let $S = \{1 \le t \le M : \mathcal{D}^{(t)}, \mathcal{E}^{(t)}\}$, where $\mathcal{D}^{(t)}$ and $\mathcal{E}^{(t)}$ are defined in Section C.1.
9:       **if** $|S| < \frac{1}{4096}MT^4$ **then**
10:          Return $\perp$.
11:       **else**
12:          By Lemma A.8, estimate the median $\widehat{\nu}$ from $\{-\widehat{\Delta}_i^{(t)}/(2.62\widehat{\Delta}_j^{(t)}) : t \in S\}$, where $\widehat{\Delta}_i^{(t)}$ and $\widehat{\Delta}_j^{(t)}$ are defined in Section C.1, and set $\widehat{\mu} = -2.62\widehat{\nu}$.
13:          **if** $|\widehat{\mu}| > 9\alpha/22$ **then**
14:             Set $E = E \sqcup \{(i, j)\}$.
15:          **end if**
16:       **end if**
17:    **end for**
18: **end for**
19: Return $E$.

---

$\mathcal{D}^{(t)}$ and $\mathcal{E}^{(t)}$, sampling $-\widehat{\Delta}_i^{(t)}/\widehat{\Delta}_j^{(t)}$ from the rest, and treating intervals that fail to satisfy $\mathcal{C}^{(t)}$ as corruption.

Let $\eta$ be an upper bound on the conditional corruption probability of an accepted interval, i.e. on $\mathbb{P}[\neg\mathcal{C}^{(t)} \mid \mathcal{D}^{(t)}, \mathcal{E}^{(t)}, s]$ for a realized start state $s$. We first bound $\eta$.

**Lemma C.8.** *We have* $\eta \le 4(1 - \exp(-2Td))$.

*Proof.* Let $s$ denote the start state of the interval. Since $P_{iiji}$ contains two distinct indices, Lemma A.3 gives

$$\mathbb{P}[\neg\mathcal{C}^{(t)}] \le 1 - e^{-2Td}.$$

Also, by Lemma A.5 and independence of the future clocks from the start state,

$$\mathbb{P}[\neg\mathcal{C}^{(t)} \mid \mathcal{D}^{(t)}, s] \le 1 - e^{-2Td}.$$

Also Lemma C.6 gives

$$\mathbb{P}[\mathcal{E}^{(t)} \mid \mathcal{D}^{(t)}, s] > 1/4.$$

Therefore, for every start state $s$,

$$\begin{aligned}
&\mathbb{P}[\neg\mathcal{C}^{(t)} \mid \mathcal{D}^{(t)}, \mathcal{E}^{(t)}, s] \\
&= \frac{\mathbb{P}[\mathcal{E}^{(t)} \mid \neg\mathcal{C}^{(t)}, \mathcal{D}^{(t)}, s]\,\mathbb{P}[\neg\mathcal{C}^{(t)} \mid \mathcal{D}^{(t)}, s]}{\mathbb{P}[\mathcal{E}^{(t)} \mid \mathcal{D}^{(t)}, s]} \\
&\le 4(1 - e^{-2Td}).
\end{aligned}$$

Averaging over the start state yields the claim. $\qquad\square$

We next show that we can collect sufficiently many samples in which $\mathcal{D}^{(t)}$ and $\mathcal{E}^{(t)}$ hold.

**Lemma C.9.** *For a Glauber dynamic of length $MT$ (split into $M$ intervals of length $T$ as above), in at least $MT^4/4096$ intervals do $\mathcal{D}^{(t)}$ and $\mathcal{E}^{(t)}$ hold with probability at least $1 - \exp(-MT^4/16384)$.*

*Proof.* Let

$$A_t := \mathbf{1}\{\mathcal{D}^{(t)} \wedge \mathcal{E}^{(t)}\},$$

and let $\mathcal{F}_t$ be the sigma-field generated by the first $t$ intervals. By Lemma A.4 applied to the pattern $P_{iiji}$,

$$\mathbb{P}[\mathcal{D}^{(t)}] \ge \frac{T^4}{512}.$$

Together with Lemma C.6, this yields

$$\mathbb{E}[A_t \mid \mathcal{F}_{t-1}] \ge \frac{T^4}{2048} =: p_0.$$

Fix $s > 0$. Since $0 \le A_t \le 1$,

$$\begin{aligned}
\mathbb{E}[e^{-sA_t} \mid \mathcal{F}_{t-1}] &= 1 - (1 - e^{-s})\mathbb{E}[A_t \mid \mathcal{F}_{t-1}] \\
&\le e^{-(1-e^{-s})p_0}.
\end{aligned}$$

Therefore

$$Z_r := \exp\left(-s\sum_{t=1}^{r} A_t + (1 - e^{-s})p_0 r\right)$$

is a supermartingale. Using Markov's inequality with $s = \log 2$,

$$\mathbb{P}\left[\sum_{t=1}^{M} A_t < \frac{Mp_0}{2}\right] \le \exp\left(\frac{\log 2}{2}Mp_0 - \frac{1}{2}Mp_0\right)$$

$$< e^{-Mp_0/8}.$$

Since $Mp_0/2 = MT^4/4096$ and $Mp_0/8 = MT^4/16384$, the claim follows. $\qquad\square$

**Putting the estimator together.**

*Proof of Lemma C.3.* First isolate the first

$$T_{\mathrm{diag}} := 4{,}000{,}000d\log(32/\delta)$$

of the trajectory. Applying Lemma B.2 with $\varepsilon = 1/10$ and failure probability $\delta/4$ to coordinates $i$ and $j$ recovers approximations $D_{ii}$ and $D_{jj}$ of the diagonal up to multiplicative factor $1 \pm 1/10$, and hence all necessary readings of $\widehat{X}_i$ and $\widehat{X}_j$, with probability at least $1 - \delta/2$. We then focus on the remainder of the trajectory.

Set

$$\phi := \frac{\alpha}{257}, \qquad T := \frac{\phi}{d}, \qquad MT^4 := \frac{128\log(8/\delta)}{\phi^2}.$$

The previous lemma gives the uniform conditional corruption bound

$$\mathbb{P}[\neg\mathcal{C}^{(t)} \mid \mathcal{D}^{(t)}, \mathcal{E}^{(t)}, s] \le 4(1 - e^{-2Td})$$
$$= 4(1 - e^{-2\phi}) < 8\phi$$

for every realized start state $s$. Set $\eta := 8\phi < 1/10$.

Since

$$\exp\left(-\frac{MT^4}{16384}\right) = \exp\left(-\frac{\log(8/\delta)}{128\phi^2}\right) < \frac{\delta}{4},$$

Lemma C.9 implies that with probability $1 - \delta/4$ the number of accepted intervals is at least

$$\frac{MT^4}{4096} = \frac{\log(8/\delta)}{32\phi^2} = \frac{2\log(8/\delta)}{\eta^2}.$$

Condition on this sample-count event, and enumerate the accepted intervals in chronological order as $t_1 < \cdots < t_N$. For $\ell = 1, \dots, N$, let

$$R_\ell := -\frac{\widehat{\Delta}_i^{(t_\ell)}}{\widehat{\Delta}_j^{(t_\ell)}}, \qquad \xi^{(\ell)} := \mathbf{1}_{\neg\mathcal{C}^{(t_\ell)}}.$$

Let $\mathcal{F}_\ell$ be the sigma-field generated by the diagonal-estimation phase together with the trajectory up to the end of interval $I_{t_\ell}$. By averaging the uniform bound above over the skipped intervals before the next accepted one, we obtain

$$\mathbb{E}[\xi^{(\ell)} \mid \mathcal{F}_{\ell-1}] \le \eta \qquad \text{for all } \ell.$$

Moreover, on the event $\xi^{(\ell)} = 0$, Lemma C.7 shows that, conditional on the full accepted interval, the clean sample $R_\ell/2.62$ has Gaussian tails dominated by

$$\mathcal{N}\left(-\frac{1}{2.62}\frac{c_j}{c_i}\Theta'_{ij}, 1\right).$$

Averaging over the accepted-interval randomness preserves these one-sided tail bounds conditional on $\mathcal{F}_{\ell-1}$. Therefore Lemma A.8 applies to the rescaled samples $R_\ell/2.62$, and with success probability $1 - \delta/4$ their sample median $\widehat{\nu}$ satisfies

$$\left|\widehat{\nu} + \frac{1}{2.62}\frac{c_j}{c_i}\Theta'_{ij}\right| < 5\eta.$$

Set $\mu' := -2.62\widehat{\nu}$. Then

$$\left|\mu' - \frac{c_j}{c_i}\Theta'_{ij}\right| < 5 \cdot 2.62 \cdot \eta < \frac{9\alpha}{22}.$$

However,

- If $i \not\sim j$, then $\Theta'_{ij} = 0$, so $|\mu'| < 9\alpha/22$.
- If $i \sim j$, then $|\Theta'_{ij}| \ge \alpha$, so

$$\left|\frac{c_j}{c_i}\Theta'_{ij}\right| \ge \frac{1 - 1/10}{1 + 1/10}\alpha > \frac{9\alpha}{11},$$

and therefore $|\mu'| > 9\alpha/22$.

Thus thresholding at $9\alpha/22$ determines whether $i \sim j$. A union bound over the diagonal-estimation phase, the sample-count event, and the robust-median step gives success probability at least $1 - \delta$.

Finally,

$$T_{\mathrm{diag}} + MT \le 4{,}000{,}000d\log(32/\delta) + \frac{128d^3\log(8/\delta)}{\phi^5}$$
$$< \frac{128 \cdot 257^5\, d^3\log(16/\delta)}{\alpha^5},$$

where the diagonal-estimation prefix is absorbed by the second term since $d \ge 1$ and $\alpha < 1$. $\qquad\square$

*Proof of Theorem C.1.* It suffices to apply Lemma C.3 with error $2\delta/n^2$ for each pair $(i, j)$. Since

$$\log\left(\frac{16}{2\delta/n^2}\right) = \log\left(\frac{8n^2}{\delta}\right) \le 2\log\left(\frac{4n}{\delta}\right),$$

the required trajectory length is bounded by the display in Theorem C.1. Then, the probability of any error among the $\binom{n}{2}$ candidate edges is less than $\delta$ by union bound. $\square$

*Proof of Corollary C.2.* We follow the proof of Lemma C.3, but estimate each diagonal entry up to multiplicative factor $1 \pm \varepsilon/20$ and replace $\alpha$ by $\alpha\varepsilon$ in the off-diagonal estimator. The resulting trajectory length is exactly the displayed bound. Report the diagonal estimates as $\widehat{\Theta}_{ii}$, and report $\widehat{\Theta}_{ij} = 0$ whenever the structure-learning step outputs $i \nsim j$.

For each pair $(i, j)$ with $i \sim j$, the proof of Lemma C.3 produces an estimate $\widehat{\mu}_{ij}$ for $\frac{c_j}{c_i}\Theta'_{ij}$ such that

$$\left|\widehat{\mu}_{ij} - \frac{c_j}{c_i}\Theta'_{ij}\right| < \frac{9\alpha\varepsilon}{22} \leq \frac{9}{22}\varepsilon|\Theta'_{ij}|,$$

since $|\Theta'_{ij}| \geq \alpha$ on edges. Also, because $c_i, c_j \in 1 \pm \varepsilon/20$,

$$\left|\frac{c_j}{c_i} - 1\right| \leq \frac{2(\varepsilon/20)}{1 - \varepsilon/20} \leq \frac{2}{19}\varepsilon.$$

Therefore

$$\left|\widehat{\mu}_{ij} - \Theta'_{ij}\right| \leq \left(\frac{9}{22} + \frac{2}{19}\right)\varepsilon|\Theta'_{ij}| = \frac{215}{418}\varepsilon|\Theta'_{ij}|.$$

Set

$$\widehat{\Theta}_{ij} := \widehat{\mu}_{ij}\sqrt{\widehat{\Theta}_{ii}\widehat{\Theta}_{jj}}.$$

Since the diagonal estimates are within a factor $1 \pm \varepsilon/20$, we have

$$\sqrt{\widehat{\Theta}_{ii}\widehat{\Theta}_{jj}} = (1 \pm \varepsilon/20)\sqrt{\Theta_{ii}\Theta_{jj}}.$$

Hence

$$\left|\widehat{\Theta}_{ij} - \Theta_{ij}\right| \leq \left(\frac{215}{418}\left(1 + \frac{1}{20}\right) + \frac{1}{20}\right)\varepsilon|\Theta_{ij}|$$
$$< 0.591\,\varepsilon|\Theta_{ij}| < \varepsilon|\Theta_{ij}|.$$

This proves the claim. $\square$

*Remark* C.10 (On large constants). The constants in this section preclude practicality. The main source is the following tradeoff. To separate the cases $\Theta'_{ij} = 0$ and $|\Theta'_{ij}| \geq \alpha$, the robust-median error after rescaling must be $O(\alpha)$. Since the hidden update corruption rate is $O(dT)$ for small window length $T$, one must take $T = O(\alpha/d)$. On the other hand, the visible pattern $(i, i, j, i)$ appears with probability $\Theta(T^4)$, so collecting the required $O(\alpha^{-2}\log(1/\delta))$ accepted windows forces a total trajectory length of order

$$\Theta\left(\frac{d^3\log(1/\delta)}{\alpha^5}\right)$$

up to constants. The concrete choice $T = \alpha/(257d)$ is simply a conservative instantiation of this tradeoff.

## D. Learning With Mixing

In this section, we give a more sample-efficient structure-learning algorithm when the mixing time is known. Unlike the mixing-free algorithm of Section C, we may now work with the shorter "$iji$" pattern. The key point is that on a clean stationary $iji$ window, the $j$-increment and the later $i$-increment form an exact bivariate Gaussian pair whose covariance is $-\Theta'_{ij}$.

Our main result in this section is as follows.

**Theorem D.1.** *Let* $0 < \alpha, \delta < 1$ *and* $n, d \in \mathbb{N}$. *Suppose we are given a Glauber trajectory evolving according to an* $(\alpha, d)$-*sparse GGM with precision matrix* $\Theta$, *and that* $t_{\mathrm{mix}}(\varepsilon)$ *is known. If the trajectory length satisfies*

$$T \geq \frac{80,000\log(2n/\delta)}{\alpha^2}\left(t_{\mathrm{mix}}\left(\frac{\alpha}{4000}\right) + \frac{8,000,000\,d^2}{\alpha^2}\right),$$

*then there is a polynomial-time algorithm that correctly outputs whether* $i \sim j$ *for each* $i$ *and* $j$ *with probability* $1 - \delta$.

Again, we fix $i$ and $j$, and evaluate the existence of the edge $i \sim j$ individually.

**Theorem D.2.** *For fixed* $i$ *and* $j$, *given a Glauber trajectory whose length satisfies*

$$T \geq \frac{40,000\log(8/\delta)}{\alpha^2}\left(t_{\mathrm{mix}}\left(\frac{\alpha}{4000}\right) + \frac{8,000,000\,d^2}{\alpha^2}\right),$$

*we may determine whether* $i \sim j$ *with probability* $1 - \delta$.

Section D.1 analyzes the statistic in the normalized model and then transfers it to the observable trajectory. Section D.2 uses these ingredients together with an epoch decomposition and mixing to build the edge test and complete the proofs of Theorems D.1 and D.2.

### D.1. Properties of the statistic

Fix $i$ and $j$. Let $T_0 \leq 1/3$, let $I_t$ denote the time interval $[t, t + T_0)$, and let $P_{iji} := (i, j, i)$. Let $\mathcal{D}^{(t)}$ denote the observable event that $I_t$ exhibits $P_{iji}$, and let $\mathcal{C}^{(t)}$ denote the event that $I_t$ is strict with respect to $P_{iji}$. By Lemma A.5, $\mathcal{D}^{(t)}$ and $\mathcal{C}^{(t)}$ are independent.

Normalize the Glauber trajectory by Theorem B.1, and let $\widehat{X}$ consist of $(1 \pm \alpha/10)$ coordinate-wise approximations of $X'$. We first analyze the normalized Glauber dynamic $X'$.

**The ideal normalized statistic.** Conditioned on $\mathcal{C}^{(t)}$ and $\mathcal{D}^{(t)}$, let $Y'^{(k)}$ denote the value of $X'$ at time $t + kT_0/3$, for $k = 0, 1, 2, 3$. Denote

$$\Delta_j'^{(t)} := Y_j'^{(2)} - Y_j'^{(0)}, \qquad \Delta_i'^{(t)} := Y_i'^{(3)} - Y_i'^{(1)}.$$

**Lemma D.3.** *Let $\theta := \Theta'_{ij}$. If the interval begins from stationarity, so that $Y'^{(0)} \sim \mathcal{N}(0, \Sigma')$, then conditioned on $\mathcal{C}^{(t)}$ and $\mathcal{D}^{(t)}$ we have*

$$\begin{pmatrix} \Delta_j'^{(t)} \\ \Delta_i'^{(t)} \end{pmatrix} \sim \mathcal{N}\left(0, \begin{pmatrix} 2 & -\theta \\ -\theta & 2 \end{pmatrix}\right).$$

*Proof.* Let $R := \Theta'Y'^{(0)}$. Since $Y'^{(0)} \sim \mathcal{N}(0, \Sigma')$ and $\Sigma' = (\Theta')^{-1}$, we have $R \sim \mathcal{N}(0, \Theta')$, with

$$\mathrm{Var}(R_i) = \mathrm{Var}(R_j) = 1$$
$$\text{and} \quad \mathrm{Cov}(R_i, R_j) = \theta.$$

Let $\varepsilon_1, \varepsilon_2, \varepsilon_3 \sim \mathcal{N}(0, 1)$ be the fresh Gaussian noises corresponding to the last update of the designated coordinate in each of the three thirds. Because $I_t$ exhibits $P_{iji}$ and is strict with respect to it, the endpoint of each prescribed coordinate is the fresh draw from the last update in that third. Hence

$$Y_i'^{(1)} = Y_i'^{(0)} - R_i + \varepsilon_1.$$

Before the middle $j$-update, the only neighbor of $j$ that may have changed is $i$, so

$$\Delta_j'^{(t)} = Y_j'^{(2)} - Y_j'^{(0)} = -R_j + \theta R_i - \theta \varepsilon_1 + \varepsilon_2.$$

Similarly, before the final $i$-update, the only neighbor of $i$ that may have changed is $j$, so

$$Y_i'^{(3)} = Y_i'^{(0)} - R_i - \theta \Delta_j'^{(t)} + \varepsilon_3,$$

and therefore

$$\Delta_i'^{(t)} = -\theta \Delta_j'^{(t)} + \varepsilon_3 - \varepsilon_1.$$

Everything is jointly Gaussian, so it remains only to compute the covariance matrix. First,

$$\mathrm{Var}(\Delta_j'^{(t)}) = \mathrm{Var}(-R_j + \theta R_i) + \mathrm{Var}(-\theta \varepsilon_1 + \varepsilon_2)$$
$$= (1 - \theta^2) + (1 + \theta^2) = 2.$$

Next,

$$\mathrm{Cov}(\Delta_i'^{(t)}, \Delta_j'^{(t)}) = -\theta \, \mathrm{Var}(\Delta_j'^{(t)}) + \mathrm{Cov}(\varepsilon_3 - \varepsilon_1, \Delta_j'^{(t)})$$
$$= -2\theta + \theta = -\theta.$$

Finally,

$$\mathrm{Var}(\Delta_i'^{(t)}) = \theta^2 \, \mathrm{Var}(\Delta_j'^{(t)}) + \mathrm{Var}(\varepsilon_3 - \varepsilon_1)$$
$$+ 2 \, \mathrm{Cov}(-\theta \Delta_j'^{(t)}, \varepsilon_3 - \varepsilon_1)$$
$$= -2\theta^2 + 2 + 2\theta^2 = 2.$$

This proves the claim. $\square$

**Corollary D.4.** *On the same clean stationary interval,*

$$\Delta_i'^{(t)} = -\frac{\Theta'_{ij}}{2} \Delta_j'^{(t)} + \zeta^{(t)},$$

*where*

$$\zeta^{(t)} \sim \mathcal{N}\left(0, 2 - \frac{(\Theta'_{ij})^2}{2}\right)$$

*and $\zeta^{(t)}$ is independent of $\Delta_j'^{(t)}$.*

*Proof.* This is the Gaussian regression formula from Lemma D.3, since

$$\frac{\mathrm{Cov}(\Delta_i'^{(t)}, \Delta_j'^{(t)})}{\mathrm{Var}(\Delta_j'^{(t)})} = -\frac{\Theta'_{ij}}{2}. \qquad \square$$

**Passing to the observable statistic.** Let $\widehat{Y}^{(k)}$ denote the value of $\widehat{X}$ at time $t + kT_0/3$, for $k = 0, 1, 2, 3$. Denote

$$\widehat{\Delta}_j^{(t)} := \widehat{Y}_j^{(2)} - \widehat{Y}_j^{(0)}, \qquad \widehat{\Delta}_i^{(t)} := \widehat{Y}_i^{(3)} - \widehat{Y}_i^{(1)}.$$

By Corollary B.7, there exist constants $c_i, c_j \in 1 \pm \alpha/10$ such that

$$\widehat{\Delta}_j^{(t)} = \frac{1}{c_j} \Delta_j'^{(t)}, \qquad \widehat{\Delta}_i^{(t)} = \frac{1}{c_i} \Delta_i'^{(t)}.$$

Hence Corollary D.4 yields

$$\widehat{\Delta}_i^{(t)} = -\frac{c_j}{2c_i} \Theta'_{ij} \widehat{\Delta}_j^{(t)} + \frac{1}{c_i} \zeta^{(t)},$$

where $\zeta^{(t)}$ is independent of $\widehat{\Delta}_j^{(t)}$ and

$$\zeta^{(t)} \sim \mathcal{N}\left(0, 2 - \frac{(\Theta'_{ij})^2}{2}\right).$$

**Conditioning on a large denominator.** Let $\mathcal{E}^{(t)}$ denote the event

$$\mathcal{E}^{(t)} := \{|\widehat{\Delta}_j^{(t)}| \geq 1.1\}.$$

**Lemma D.5.** *We have*

$$\mathbb{P}[\mathcal{E}^{(t)} \mid \mathcal{D}^{(t)}, \widehat{Y}^{(0)}] > \frac{2}{9}.$$

*Proof.* Since $c_j < 1.1$, the event $|\widehat{\Delta}_j^{(t)}| \geq 1.1$ is implied by $|\Delta_j'^{(t)}| \geq 1.21$. Let $Z$ be the value of $X'$ right before the last update of $j$ inside the middle third. Conditioned on $Z$, the endpoint $\Delta_j'^{(t)}$ is Gaussian with variance 1, say $\Delta_j'^{(t)} \mid Z \sim \mathcal{N}(m, 1)$ for some $m$.

Let

$$f(m) := \mathbb{P}_{x \sim \mathcal{N}(m, 1)}[|x| \leq 1.21].$$

Then $f$ is even, and for $m \geq 0$,

$$f'(m) = \varphi(1.21 + m) - \varphi(1.21 - m) \leq 0.$$

Therefore $f$ is maximized at $m = 0$, so

$$\mathbb{P}[|\Delta_j'^{(t)}| \geq 1.21 \mid Z] \geq \mathbb{P}_{x \sim \mathcal{N}(0,1)}[|x| \geq 1.21] > \frac{2}{9}.$$

Averaging over $Z$ proves the claim. $\qquad\square$

**Lemma D.6.** *Condition on a clean stationary interval and on $\mathcal{E}^{(t)}$. Then*

$$-2\frac{\widehat{\Delta}_i^{(t)}}{\widehat{\Delta}_j^{(t)}} = \frac{c_j}{c_i}\Theta'_{ij} + \sigma^{(t)}\varepsilon^{(t)},$$

*where $\varepsilon^{(t)} \sim \mathcal{N}(0,1)$, $\sigma^{(t)}$ is measurable with respect to $\widehat{\Delta}_j^{(t)}$, and*

$$|\sigma^{(t)}| < 2.86.$$

*Proof.* By the regression identity above,

$$-2\frac{\widehat{\Delta}_i^{(t)}}{\widehat{\Delta}_j^{(t)}} = \frac{c_j}{c_i}\Theta'_{ij} - 2\frac{\zeta^{(t)}}{c_i\widehat{\Delta}_j^{(t)}}.$$

Since $\zeta^{(t)}$ is independent of $\widehat{\Delta}_j^{(t)}$ and has variance at most 2, the second term equals $\sigma^{(t)}\varepsilon^{(t)}$ for some standard Gaussian $\varepsilon^{(t)}$ and

$$|\sigma^{(t)}| \leq \frac{2\sqrt{2}}{|c_i\widehat{\Delta}_j^{(t)}|} < \frac{2\sqrt{2}}{0.9 \cdot 1.1} < 2.86,$$

where we use $\mathcal{E}^{(t)}$. $\qquad\square$

### D.2. Edge detection

We divide the trajectory into $M$ *epochs*, each consisting of a mixing period of length $t_{\mathrm{mix}}(\beta)$ followed by $N$ candidate intervals of length $T_0$, where

$$T_0 = \frac{\alpha}{400d}, \quad \beta = \frac{\alpha}{4000}, \quad N = \frac{50}{T_0^3}.$$

In the $k$th epoch, let $L_k$ denote the first candidate interval for which $\mathcal{D}^{(t)}$ holds. If no such interval exists, the epoch is discarded. If $L_k$ exists but $\mathcal{E}^{(t)}$ fails on that interval, the epoch is also discarded. Otherwise we record the sample

$$R_k := -2\frac{\widehat{\Delta}_i^{(k)}}{\widehat{\Delta}_j^{(k)}},$$

where $\widehat{\Delta}_i^{(k)}$ and $\widehat{\Delta}_j^{(k)}$ are computed from the selected interval $L_k$.

The resulting procedure is given by Algorithm 3.

We now analyze the estimator.

---

**Algorithm 3** $E = \text{LearnGGMMixed}(n, d, \alpha, \delta, t_{\mathrm{mix}})$

1: Set $T_0 = \alpha/(400d)$, $\beta = \alpha/4000$, $N = 50/T_0^3$, and $M = 80{,}000 \log(2n/\delta)/\alpha^2$.
2: Let $\mathrm{diag}(\widetilde{\Theta}) = \text{EstimateDiagonal}(n, d, \alpha, \alpha/10, \delta/2)$ from Algorithm 1.
3: Keep the unused part of the trajectory.
4: Let $E = \varnothing$.
5: Observe a trajectory of length $M(t_{\mathrm{mix}}(\beta) + NT_0)$, split into $M$ epochs.
6: **for** $i = 1, \ldots, n$ **do**
7:    **for** $j = i + 1, \ldots, n$ **do**
8:       Let $S = \varnothing$.
9:       **for** $k = 1, \ldots, M$ **do**
10:          After the mixing segment of epoch $k$, split the remainder into candidate intervals $I_{k,1}, \ldots, I_{k,N}$ of length $T_0$.
11:          Let $L_k$ be the first interval for which $\mathcal{D}^{(t)}$ holds.
12:          **if** $L_k$ exists and $\mathcal{E}^{(t)}$ holds on $L_k$ **then**
13:             Record $R_k = -2\widehat{\Delta}_i^{(k)}/\widehat{\Delta}_j^{(k)}$ and append it to $S$.
14:          **end if**
15:       **end for**
16:       **if** $|S| < M/16$ **then**
17:          Return $\perp$.
18:       **else**
19:          Let $\widehat{\mu}$ be the sample median of $S$.
20:          **if** $|\widehat{\mu}| > 9\alpha/22$ **then**
21:             Set $E = E \sqcup \{(i, j)\}$.
22:          **end if**
23:       **end if**
24:    **end for**
25: **end for**
26: Return $E$.

---

**Lemma D.7.** *Let $A_k$ denote the indicator that epoch $k$ records a sample. With the choice $N = 50/T_0^3$, each epoch records a sample with conditional probability at least $1/8$ given the past. Consequently,*

$$\mathbb{P}\left[\sum_{k=1}^{M} A_k < \frac{M}{16}\right] \leq e^{-M/128}.$$

*Proof.* Fix an epoch and condition on the past. For a single candidate interval, Lemma A.4 gives

$$\mathbb{P}[\mathcal{D}^{(t)}] \geq \frac{T_0^3}{54}.$$

Therefore the probability that some candidate interval satisfies $\mathcal{D}^{(t)}$ is at least

$$1 - \left(1 - \frac{T_0^3}{54}\right)^N \geq 1 - e^{-50/54} > \frac{3}{5}.$$

Let $L_k$ be the first such interval. Conditioned on $L_k$ and on the past, Lemma D.5 gives

$$\mathbb{P}[\mathcal{E}^{(t)} \text{ holds on } L_k \mid L_k, \text{ past}] > \frac{2}{9}.$$

Therefore the conditional probability that the epoch records a sample is at least

$$\frac{3}{5} \cdot \frac{2}{9} = \frac{2}{15} > \frac{1}{8}.$$

Let $\mathcal{F}_k$ be the sigma-field generated by the first $k$ epochs, and define

$$M_r := \sum_{k=1}^{r} (A_k - \mathbb{E}[A_k \mid \mathcal{F}_{k-1}]).$$

Then $(M_r, \mathcal{F}_r)$ is a martingale with increments in $[-1, 1]$, and $\sum_{k=1}^{M} \mathbb{E}[A_k \mid \mathcal{F}_{k-1}] \geq M/8$. Thus

$$\mathbb{P}\left[\sum_{k=1}^{M} A_k < \frac{M}{16}\right] \leq \mathbb{P}\left[M_M < -\frac{M}{16}\right] \leq e^{-M/128}$$

by Azuma–Hoeffding. □

**Lemma D.8.** *Suppose*

$$T_0 = \frac{\alpha}{400d} \quad and \quad \beta = \frac{\alpha}{4000}.$$

*Then, conditional on the past and on the event that an epoch records a sample, that sample is corrupted with probability less than $\alpha/35$.*

*Proof.* There are two sources of corruption.

First, non-strict pattern intervals. Since future clocks are independent of the past and $P_{iji}$ contains two distinct indices, Lemmas A.3 and A.5 give

$$\mathbb{P}[\neg\mathcal{C}^{(t)} \mid \mathcal{D}^{(t)}] \leq 1 - e^{-2dT_0}.$$

By Lemma D.5, conditioning additionally on $\mathcal{E}^{(t)}$ inflates this by at most a factor of $9/2$, so the non-strictness contribution to the conditional corruption probability is at most

$$\eta_{\text{strict}} \leq \frac{1 - e^{-2dT_0}}{2/9} < 9dT_0 = \frac{9\alpha}{400}.$$

Second, imperfect mixing. At the start of each epoch, after waiting $t_{\text{mix}}(\beta)$, the epoch-start distribution is within TV-distance $\beta$ of stationarity. By Corollary A.15, the same is true at the start of any later candidate interval in the epoch. A maximal coupling with stationarity therefore fails with probability at most $\beta$. Since an epoch records a sample with conditional probability at least $1/8$ by Lemma D.7, conditioning on the epoch being recorded inflates this by at most a factor of $8$. Thus the mixing contribution to corruption is at most

$$\eta_{\text{mix}} \leq 8\beta = \frac{\alpha}{500}.$$

Combining the two bounds gives

$$\eta \leq \eta_{\text{strict}} + \eta_{\text{mix}} < \frac{9\alpha}{400} + \frac{\alpha}{500} = \frac{49\alpha}{2000} < \frac{\alpha}{35}. \quad □$$

**Putting the edge test together.**

*Proof of Theorem D.2.* First isolate the initial

$$T_{\text{diag}} := \frac{4{,}000{,}000d \log(32/\delta)}{\alpha^3}$$

portion of the trajectory, and apply Lemma B.2 to retrieve approximations of the diagonal up to multiplicative factor $1 \pm \alpha/10$, and hence all necessary readings of $\widehat{X}_i$ and $\widehat{X}_j$, with probability at least $1 - \delta/2$.

Now take

$$T_0 = \frac{\alpha}{400d}, \quad \beta = \frac{\alpha}{4000}, \quad N = \frac{50}{T_0^3},$$

$$\text{and} \quad M = \frac{40{,}000 \log(8/\delta)}{\alpha^2}.$$

By Lemma D.7, with probability at least $1 - \delta/4$, the number of recorded samples is at least

$$\frac{M}{16} = \frac{2500 \log(8/\delta)}{\alpha^2}.$$

Let $k_1 < \cdots < k_N$ denote the recorded epochs in chronological order, and condition on the sample-count event from Lemma D.7, so that

$$N \geq \frac{M}{16} = \frac{2500 \log(8/\delta)}{\alpha^2}.$$

For $\ell = 1, \ldots, N$, write

$$R_\ell := R_{k_\ell}, \qquad \xi^{(\ell)} := \mathbf{1}_{\{\text{epoch } k_\ell \text{ is corrupted}\}}.$$

Let $\mathcal{F}_\ell$ be the sigma-field generated by the trajectory up to the end of epoch $k_\ell$. By averaging Lemma D.8 over the skipped epochs before the next recorded one, we obtain

$$\mathbb{E}[\xi^{(\ell)} \mid \mathcal{F}_{\ell-1}] \leq \eta \qquad \text{for all } \ell,$$

with

$$\eta < \frac{\alpha}{35}.$$

On the event $\xi^{(\ell)} = 0$, Lemma D.6 implies that, conditional on $\mathcal{F}_{\ell-1}$, the clean sample $R_\ell/2.86$ has Gaussian tails dominated by

$$\mathcal{N}\left(\frac{1}{2.86} \frac{c_j}{c_i} \Theta'_{ij}, 1\right).$$

Let

$$\mu := \frac{c_j}{c_i} \Theta'_{ij}.$$

Since

$$\frac{M}{16} = \frac{2500 \log(8/\delta)}{\alpha^2} > \frac{2 \log(8/\delta)}{(\alpha/35)^2},$$

Lemma A.8 applies to the rescaled samples $R_\ell/2.86$. Therefore, with probability at least $1 - \delta/4$, the sample median $\widehat{\mu}$ satisfies

$$|\widehat{\mu} - \mu| < 5 \cdot 2.86 \cdot \frac{\alpha}{35} < \frac{9\alpha}{22}.$$

Now:

- If $i \nsim j$, then $\Theta'_{ij} = 0$, so $\mu = 0$.

- If $i \sim j$, then $|\Theta'_{ij}| \geq \alpha$, and since $c_i, c_j \in 1 \pm \alpha/10$,

$$|\mu| = \left|\frac{c_j}{c_i} \Theta'_{ij}\right| \geq \frac{1 - \alpha/10}{1 + \alpha/10} \alpha > \frac{9\alpha}{11}.$$

Hence thresholding at $9\alpha/22$ separates the two cases.

A final union bound over the diagonal-estimation phase, the sample-count event, and the robust-median step gives success probability at least $1 - \delta$, since each individual event has failure probability upper bounded by $\frac{\delta}{4}$.

The per-epoch non-mixing cost is

$$NT_0 = \frac{50}{T_0^2} = 50\left(\frac{400d}{\alpha}\right)^2 = \frac{8{,}000{,}000\, d^2}{\alpha^2}.$$

Therefore the total trajectory length is at most

$$\begin{aligned}
&T_{\text{diag}} + M\big(t_{\text{mix}}(\beta) + NT_0\big) \\
&\leq \frac{40{,}000 \log(8/\delta)}{\alpha^2}\left(t_{\text{mix}}\left(\frac{\alpha}{4000}\right) + \frac{8{,}000{,}000\, d^2}{\alpha^2}\right),
\end{aligned}$$

where the diagonal-estimation prefix is absorbed by the $d^2/\alpha^4$ term since $d \geq 1$ and $\alpha < 1$. $\qquad \square$

*Proof of Theorem D.1.* Apply Theorem D.2 with error $2\delta/n^2$ for each pair $(i, j)$. Since

$$\log\left(\frac{8}{2\delta/n^2}\right) = \log\left(\frac{4n^2}{\delta}\right) \leq 2\log\left(\frac{2n}{\delta}\right),$$

the required trajectory length is bounded by the display in Theorem D.1. A union bound over the $\binom{n}{2}$ candidate edges now gives success probability at least $1 - \delta$. $\qquad \square$

# E. Information-Theoretic Lower Bound

In this section, we prove the lower bound by constructing a hard family of graphs, upper-bounding the pairwise KL divergence between the resulting trajectory distributions, and applying Fano's inequality. By Section G, this yields the corresponding continuous-time lower bound up to the usual factor of $n$.

Our strategy is to decompose the KL divergence step-wise. And upper bound the expected KL divergence introduced at each update, using the fact that KL divergence between Gaussians with the same variance is proportional to the square of the mean difference. Our initial decomposition is the same as that of (Tirukkonda et al., 2025), but instead of upper bounding the KL divergence by the worst case update, our analysis involving the average case mean difference produces a $\log(n)$ improvement over the previous result.

We formalize recovery through graph-learning tests:

**Definition E.1.** A *graph-learning test* is a function that takes as input a Glauber trajectory $\mathcal{T}$ generated from an $(\alpha, d)$-GGM with precision matrix $\Theta$ and support graph $G$, together with the values $\alpha$ and $d$, and outputs an estimate $\widehat{G}$ for $G$. Its success event is $\widehat{G} = G$.

From now on, we consider only well-mixed Glauber trajectories:

**Definition E.2.** Let $\mathcal{T}(N, \Theta)$ denote a Glauber trajectory, with $N$ updates, with precision matrix $\Theta$, and with starting point sampled from $\mathcal{N}(0, \Theta^{-1})$.

Our main result in this section is as follows:

**Theorem E.3.** *Let $n \geq 16$ and $N$ be positive integers, and let $0 < \alpha < 1/4$. There exists a set $S$ of $2n$-dimensional $(\alpha, 1)$-sparse GGMs such that, if $\Theta$ is sampled uniformly*

*from $S$ and the trajectory $\mathcal{T}(N, \Theta)$ is generated therefrom, then the success rate of a graph-learning test is at most $1/2$ whenever $N \leq \frac{n \log n}{8\alpha^2}$.*

Note that $(\alpha, 1)$-sparse GGMs are also $(\alpha, d)$-sparse for every positive integer $d$, so this bound holds for all $d$.

### E.1. The class of graphs

Fix $n$ a positive integer, and consider the graph $G_0$ of sparsity 1 on vertices labeled 1, 2, …, $2n$, with an edge $(2k - 1) \sim 2k$ for each $k = 1, \ldots, n$, and no other edges. Further, let $G_k$ be a copy of $G_0$, but with the edge $(2k - 1) \sim 2k$ removed, for $k = 1, \ldots, n$.

For each graph $G_k$, define $\Theta_k$ as the following $(\alpha, 1)$-sparse GGM with underlying graph $G_k$:

$$
(\Theta_k)_{ij} = \begin{cases} 1 & i = j \\ \alpha & i \sim j \text{ in } G_k \\ 0 & \text{else.} \end{cases}
$$

Our set of GGMs is $S = \{\Theta_1, \ldots, \Theta_n\}$.

Notice that each GGM is the direct sum of $2 \times 2$ matrices, among which at least one is the identity matrix $I_2$, and the remaining are

$$
M_\alpha = \begin{bmatrix} 1 & \alpha \\ \alpha & 1 \end{bmatrix}.
$$

The eigenvalues of $M_\alpha$ are $1 - \alpha > 0$ and $1 + \alpha > 0$, so all $\Theta_k$ are positive definite.

### E.2. A bound on KL-divergence

We now upper-bound the pairwise KL divergence between the candidate trajectory distributions. For a state $X$ before the $i$th update, write $\mathcal{T}(i, \Theta_r)_{X^{(i)}|X^{(i-1)}=X}$ for the conditional distribution of the post-update state under $\Theta_r$, and $\mathcal{T}(i, \Theta_r)_{X_\ell^{(i)}|X^{(i-1)}=X}$ for its $\ell$th coordinate, where $r \in \{0, 1\}$.

**Lemma E.4.** *For each $k = 1, \ldots, n$, we have*

$$
D_{\mathrm{KL}}(\mathcal{T}(N, \Theta_k) \| \mathcal{T}(N, \Theta_0)) \leq \left(\frac{N}{n} + 1\right)\alpha^2.
$$

*Proof of Lemma E.4.* Without loss of generality $k = 1$. Recall both trajectories $\mathcal{T}(N, \Theta_0)$ and $\mathcal{T}(N, \Theta_k)$ begin mixed, so we first compute the KL-divergence of their initial positions, by

$$
\begin{aligned}
& D_{\mathrm{KL}}(\mathcal{T}(0, \Theta_1)_{X^{(0)}} \| \mathcal{T}(0, \Theta_0)_{X^{(0)}}) \\
&= D_{\mathrm{KL}}(\mathcal{N}(0, \Theta_1^{-1}) \| \mathcal{N}(0, \Theta_0^{-1})) \\
&= D_{\mathrm{KL}}(\mathcal{N}(0, M_\alpha) \| \mathcal{N}(0, I_2)) \\
&= -\frac{1}{2}\ln(1 - \alpha^2) < \alpha^2,
\end{aligned}
$$

since $\alpha < 1/4$.

Now, by coupling, we may assume the two trajectories share the same sequence of updates. Suppose both trajectories have reached position $X$ before the $i$th update, and on the $i$th update, both trajectories update the $\ell$th coordinate. We explicitly write down the distribution of the $\ell$th coordinate after this Glauber update:

$$
\mathcal{T}(i, \Theta_0)_{X_\ell^{(i)}|X^{(i-1)}=X} = -\sum_{j \sim \ell} -(\Theta_0)_{j\ell}X_j + \mathcal{N}(0, 1),
$$

$$
\mathcal{T}(i, \Theta_1)_{X_\ell^{(i)}|X^{(i-1)}=X} = -\sum_{j \sim \ell} -(\Theta_1)_{j\ell}X_j + \mathcal{N}(0, 1)
$$

But recall $(\Theta_0)_{j\ell} = (\Theta_1)_{j\ell}$ unless $\{j, \ell\} = \{1, 2\}$, in which case $(\Theta_0)_{12} - (\Theta_1)_{12} = \alpha$. Hence, the two distributions are normal with the same variance, and the means differ by $\alpha X_2$ if $\ell = 1$, by $\alpha X_1$ if $\ell = 2$, and by 0 otherwise.

It follows that the KL-divergence for the $i$th update is given by

$$
\begin{aligned}
& D_{\mathrm{KL}}\left(\mathcal{T}(i, \Theta_1)_{X^{(i)}|X^{(i-1)}=X} \| \mathcal{T}(i, \Theta_0)_{X^{(i)}|X^{(i-1)}=X}\right) \\
&= D_{\mathrm{KL}}\left(\mathcal{T}(i, \Theta_1)_{X_\ell^{(i)}|X^{(i-1)}=X} \right. \\
& \qquad \left. \| \mathcal{T}(i, \Theta_0)_{X_\ell^{(i)}|X^{(i-1)}=X}\right) \\
&= \mathbb{E}_\ell\left[\begin{cases} \frac{1}{2}\alpha^2 X_2^2 & \ell = 1 \\ \frac{1}{2}\alpha^2 X_1^2 & \ell = 2 \\ 0 & \ell \geq 3 \end{cases}\right] = \frac{\alpha^2}{4n}\left(X_1^2 + X_2^2\right).
\end{aligned}
$$

But by Lemma A.12, we always have $X \sim \mathcal{N}(0, \Theta_1^{-1})$, so $\mathbb{E}_X\left[X_1^2\right] = (\Theta_1^{-1})_{11} = \frac{1}{1-\alpha^2} < 2$, and thus

$$
\begin{aligned}
& \mathbb{E}_{X^{(i-1)}}\left[D_{\mathrm{KL}}\left(\mathcal{T}(i, \Theta_1)_{X^{(i)}|X^{(i-1)}=X}\right.\right. \\
& \qquad\qquad \left.\left. \| \mathcal{T}(i, \Theta_0)_{X^{(i)}|X^{(i-1)}=X}\right)\right] \\
&= \frac{\alpha^2}{2n(1-\alpha^2)} < \frac{\alpha^2}{n}.
\end{aligned}
$$

We conclude by linearity of expectation,

$$
\begin{aligned}
& D_{\mathrm{KL}}(\mathcal{T}(N, \Theta_1) \| \mathcal{T}(N, \Theta_0)) \\
&= D_{\mathrm{KL}}\left(\mathcal{T}(0, \Theta_1)_{X^{(0)}} \| \mathcal{T}(0, \Theta_0)_{X^{(0)}}\right) \\
& \quad + \sum_{i=1}^N \mathbb{E}_{X^{(i-1)}}\left[D_{\mathrm{KL}}\left(\mathcal{T}(i, \Theta_1)_{X^{(i)}|X^{(i-1)}=X}\right.\right. \\
& \qquad\qquad\qquad \left.\left. \| \mathcal{T}(i, \Theta_0)_{X^{(i)}|X^{(i-1)}=X}\right)\right] \\
&< \alpha^2 + \frac{N\alpha^2}{n}. \qquad\qquad\qquad\qquad\qquad \square
\end{aligned}
$$

### E.3. Fano's method

We finish by using Fano's method to bound the success rate of a graph-learning test.

**Lemma E.5.** *The success rate of a graph-learning test is bounded by*

$$\mathbb{P}\left[\widehat{G} = G\right] \leq \frac{(N+n)\alpha^2}{n \log n} + \frac{\log 2}{\log n}.$$

*Proof.* Let $V$ be sampled uniformly from $\{1, \ldots, n\}$. Let $\mathcal{T}$ denote the observed trajectory. We may then bound the mutual information between $V$ and the observed trajectory by

$$I(V; \mathcal{T}) \leq \frac{1}{n} \sum_{i=1}^{n} D_{\mathrm{KL}}(\mathcal{T}(N, \Theta_i) \| \mathcal{T}(N, \Theta_0))$$
$$< \alpha^2 + \frac{N\alpha^2}{n}.$$

By Fano's inequality, we have

$$\mathbb{P}\left[\widehat{G} = G\right] \leq \frac{I(V; \mathcal{T}) + \log 2}{\log n}$$
$$\leq \frac{(N+n)\alpha^2}{n \log n} + \frac{\log 2}{\log n}. \qquad \square$$

*Proof of Theorem E.3.* With our given assumptions, we have $N + n \leq \frac{n \log n}{4\alpha^2}$, and so by Lemma E.5, we have

$$\mathbb{P}\left[\widehat{G} = G\right] \leq \frac{(N+n)\alpha^2}{n \log n} + \frac{\log 2}{\log n}$$
$$\leq \frac{1}{4} + \frac{1}{4} = \frac{1}{2}. \qquad \square$$

## F. Robust Estimators

In this appendix, we prove the two robust-estimation lemmas stated in Section A, namely Lemmas A.7 and A.8. The variance proof is based on controlling the relevant quantile under contamination, while the mean proof uses a martingale argument together with Azuma–Hoeffding.

**Lemma A.7** (Robust variance estimation)**.** *Let $0 < \eta \leq 1/10$ and $0 < \delta < 1$. There is a linear-time algorithm that takes $n$ samples from $\mathcal{N}(0, \sigma^2)$, an $\eta$-fraction of which are corrupted (as in Definition A.6), and outputs $\widehat{\sigma}$ such that $|\widehat{\sigma} - \sigma| < 5\eta\sigma$ with probability $1 - \delta$, provided $n \geq \frac{2 \log(4/\delta)}{\eta^2}$.*

*Proof.* Let $\Phi$ and $\varphi$ denote the cdf and pdf, respectively, of the standard Gaussian, and set

$$q := \Phi^{-1}(3/4).$$

Let $M$ be the sample median of the absolute values of the observed samples, and set

$$\widehat{\sigma} := \frac{M}{q}.$$

By scaling, it suffices to consider the case $\sigma = 1$. Then the absolute value of an uncorrupted sample has cdf

$$G(t) := 2\Phi(t) - 1.$$

Let $G_n$ denote the empirical cdf of the clean absolute values, let $H_n$ denote the empirical cdf of the observed absolute values, and let

$$k := \sum_{i=1}^{n} \xi_i$$

be the number of corruptions.

By Dvoretzky–Kiefer–Wolfowitz with tolerance $\eta/2$,

$$\mathbb{P}\left[\sup_{x \in \mathbb{R}} |G_n(x) - G(x)| > \frac{\eta}{2}\right] \leq 2e^{-n\eta^2/2} \leq \frac{\delta}{2}.$$

Also, a multiplicative Chernoff bound with parameter $1/3$ gives

$$\mathbb{P}\left[k > \frac{4}{3}\eta n\right] \leq \exp\left(-\left(\frac{4}{3} \log \frac{4}{3} - \frac{1}{3}\right)\eta n\right)$$
$$\leq e^{-\eta n/20} \leq \frac{\delta}{4},$$

where the last inequality uses

$$n \geq \frac{2 \log(4/\delta)}{\eta^2} \quad \text{and} \quad \eta \leq \frac{1}{10}.$$

Therefore, with probability at least $1 - 3\delta/4$,

$$\sup_{x \in \mathbb{R}} |G_n(x) - G(x)| \leq \frac{\eta}{2} \quad \text{and} \quad k \leq \frac{4}{3}\eta n.$$

Work on this event. Since $H_n$ is obtained from $G_n$ by changing at most $k$ sample points,

$$\sup_{x \in \mathbb{R}} |H_n(x) - G(x)|$$
$$\leq \sup_{x \in \mathbb{R}} |H_n(x) - G_n(x)| + \sup_{x \in \mathbb{R}} |G_n(x) - G(x)|$$
$$\leq \frac{k}{n} + \frac{\eta}{2} \leq \frac{11\eta}{6}.$$

Since $M$ is a median of the observed absolute values, we have $H_n(M^-) \leq 1/2 \leq H_n(M)$. Because $G$ is continuous and increasing, it follows that

$$\frac{1}{2} - \frac{11\eta}{6} \leq G(M) \leq \frac{1}{2} + \frac{11\eta}{6}.$$

Equivalently,

$$\frac{3}{4} - \frac{11\eta}{12} \leq \Phi(M) \leq \frac{3}{4} + \frac{11\eta}{12}.$$

Since $\eta \leq 1/10$, this places $M$ in the interval

$$\Phi^{-1}\left(\frac{79}{120}\right) \leq M \leq \Phi^{-1}\left(\frac{101}{120}\right).$$

Now $\Phi^{-1}$ is increasing and convex on $(1/2, 1)$, so its deviation from $q = \Phi^{-1}(3/4)$ is bounded linearly on this interval. Because

$$\frac{11\eta}{12} \leq \frac{11}{120} \cdot 10\eta,$$

we obtain

$$|M - q| \leq 10\eta \cdot \max\left\{\Phi^{-1}\left(\frac{101}{120}\right) - q,\right.$$
$$\left. q - \Phi^{-1}\left(\frac{79}{120}\right)\right\}.$$

Finally,

$$|\widehat{\sigma} - 1| = \frac{|M - q|}{q}$$
$$\leq \frac{10\eta}{\Phi^{-1}(3/4)} \max\left\{\Phi^{-1}\left(\frac{101}{120}\right) - \Phi^{-1}\left(\frac{3}{4}\right),\right.$$
$$\left. \Phi^{-1}\left(\frac{3}{4}\right) - \Phi^{-1}\left(\frac{79}{120}\right)\right\}$$
$$< 5\eta.$$

This proves the lemma. $\qquad\square$

**Lemma A.8** (Robust mean estimation). *Let $0 < \eta \leq 1/10$ and $0 < \delta < 1$. Let $\Phi$ denote the cdf of a standard Gaussian. We are given a filtration $(\mathcal{F}_\ell)_{\ell=0}^{n}$ and samples $x^{(1)}, \ldots, x^{(n)}$. Suppose there exist indicators $\xi^{(\ell)} \in \{0, 1\}$ such that*

$$\mathbb{E}[\xi^{(\ell)} \mid \mathcal{F}_{\ell-1}] \leq \eta \qquad \text{for all } \ell.$$

*Assume moreover that whenever $\xi^{(\ell)} = 0$, the clean sample is centered at the same value $\mu$ and has Gaussian tails dominated by unit variance in the sense that, for every $t \geq 0$,*

$$\mathbb{P}\left[x^{(\ell)} \geq \mu + t \;\middle|\; \mathcal{F}_{\ell-1}, \xi^{(\ell)} = 0\right] \leq 1 - \Phi(t),$$

$$\mathbb{P}\left[x^{(\ell)} \leq \mu - t \;\middle|\; \mathcal{F}_{\ell-1}, \xi^{(\ell)} = 0\right] \leq 1 - \Phi(t).$$

*Then the sample median $\widehat{\mu}$ satisfies $|\widehat{\mu} - \mu| < 5\eta$ with probability $1 - \delta$, provided $n \geq \frac{2\log(2/\delta)}{\eta^2}$.*

*Proof.* Let $(\mathcal{F}_\ell)_{\ell=0}^{n}$ be the filtration from the statement, and let $\widehat{\mu}$ denote the sample median. Let $\Phi$ and $\varphi$ denote the cdf and pdf, respectively, of a standard Gaussian. Let $\mathbb{1}_\ell$

indicate the event that either $\xi^{(\ell)} = 1$ or $x^{(\ell)} > \mu + 5\eta$. Then

$$\mathbb{E}[\mathbb{1}_\ell \mid \mathcal{F}_{\ell-1}]$$
$$\leq \mathbb{E}[\xi^{(\ell)} \mid \mathcal{F}_{\ell-1}] + \mathbb{P}\left[x^{(\ell)} > \mu + 5\eta, \; \xi^{(\ell)} = 0 \;\middle|\; \mathcal{F}_{\ell-1}\right]$$
$$\leq \eta + \mathbb{P}\left[x^{(\ell)} > \mu + 5\eta \;\middle|\; \mathcal{F}_{\ell-1}, \xi^{(\ell)} = 0\right]$$
$$\leq 1 + \eta - \Phi(5\eta).$$

Therefore,

$$M_k = \sum_{\ell=1}^{k} \left(\mathbb{1}_\ell - \mathbb{E}[\mathbb{1}_\ell \mid \mathcal{F}_{\ell-1}]\right)$$

is a martingale with bounded increments in $[-1, 1]$.

By Azuma–Hoeffding (Fact A.10),

$$\mathbb{P}[\widehat{\mu} \geq \mu + 5\eta] \leq \mathbb{P}\left[\sum_{\ell=1}^{n} \mathbb{1}_\ell \geq \frac{n}{2}\right]$$
$$\leq \mathbb{P}\left[M_n \geq n\left(\Phi(5\eta) - \eta - \frac{1}{2}\right)\right]$$
$$\leq \exp\left(-2n\left(\Phi(5\eta) - \eta - \frac{1}{2}\right)^2\right).$$

For $\eta \leq 1/10$, we have $\varphi(5\eta) > 3/10$, and hence

$$\Phi(5\eta) - \eta - \frac{1}{2} = -\eta + \int_0^{5\eta} \varphi(x)\, dx$$
$$> -\eta + 5\eta\varphi(5\eta) > \eta/2.$$

Substituting this bound yields

$$\mathbb{P}[\widehat{\mu} \geq \mu + 5\eta] \leq \exp(-n\eta^2/2).$$

The lower tail is identical: if $\widetilde{\mathbb{1}}_\ell$ denotes the event that either $\xi^{(\ell)} = 1$ or $x^{(\ell)} < \mu - 5\eta$, the same argument gives

$$\mathbb{P}[\widehat{\mu} \leq \mu - 5\eta] \leq \exp(-n\eta^2/2).$$

Combining the two one-sided bounds with a union bound, we obtain

$$\mathbb{P}[|\widehat{\mu} - \mu| \geq 5\eta] \leq 2\exp(-n\eta^2/2) \leq \delta.$$

This proves the lemma. $\qquad\square$

## G. Continuous time versus number of updates

The upper bounds in Theorems 1.3 and 1.4 are stated in terms of the continuous-time horizon $T$, whereas the lower bound in Theorem E.3 is stated in terms of the number of

single-site updates. This appendix records the standard reduction showing that these two formulations are equivalent up to the factor $n$.

Recall from Definition 1.2 that the jump times satisfy $S^{(0)} = 0$ and $S^{(\ell+1)} - S^{(\ell)} \sim \text{Exp}(n)$ i.i.d. Let

$$N_T := \max\{\ell \geq 0 : S^{(\ell)} \leq T\}$$

denote the number of updates by time $T$. Then $N_T \sim \text{Poisson}(nT)$. Moreover, conditional on $N_T = m$, the continuous-time trajectory up to time $T$ is exactly the first $m$ steps of the embedded discrete-time chain, together with the jump times $S^{(1)}, \ldots, S^{(m)}$. Since these jump times are independent of $\Theta$ and of the embedded chain, the only substantive difference between the two models is that in continuous time one observes a *random* number $N_T$ of discrete-time updates.

We will use the following standard Chernoff bounds for a Poisson random variable: for every $\lambda > 0$,

$$\mathbb{P}\big[\text{Poisson}(2\lambda) < \lambda\big] \leq e^{-\lambda/4}$$
$$\text{and} \quad \mathbb{P}\big[\text{Poisson}(\lambda) > 2\lambda\big] \leq e^{-\lambda/3}.$$

**Proposition G.1.** *The continuous-time and discrete-time formulations are interchangeable up to constant factors.*

(i) *If there is an estimator that, from the first $N$ updates of the embedded discrete-time chain, succeeds with probability at least $1 - \delta$, then there is an estimator that, from a continuous-time trajectory of length $2N/n$, succeeds with probability at least $1 - \delta - e^{-N/4}$.*

(ii) *If there is an estimator that, from a continuous-time trajectory of length $T$, succeeds with probability at least $1 - \delta$, then there is an estimator that, from the first $\lceil 2nT \rceil$ updates of the embedded discrete-time chain, succeeds with probability at least $1 - \delta - e^{-nT/3}$.*

*Proof.* For (i), observe the continuous-time trajectory up to time $2N/n$. If $N_{2N/n} \geq N$, run the discrete-time estimator on the first $N$ updates of the embedded chain and ignore the rest. Since $N_{2N/n} \sim \text{Poisson}(2N)$, the bad event is $\{N_{2N/n} < N\}$, which has probability at most $e^{-N/4}$.

For (ii), suppose we are given the first $M = \lceil 2nT \rceil$ updates of the embedded discrete-time chain. Sample i.i.d. waiting times $W_1, \ldots, W_M \sim \text{Exp}(n)$, set $S^{(\ell)} = \sum_{r=1}^{\ell} W_r$, and reconstruct the corresponding continuous-time path up to time $T$. If $S^{(M)} > T$, this reconstruction is exact up to time $T$, so we may run the continuous-time estimator. The bad event is $S^{(M)} \leq T$, equivalently $N_T \geq M$ for a rate-$n$ Poisson process, and since $M \geq 2nT$ this has probability at most $e^{-nT/3}$. $\qquad\square$

**Corollary G.2.** *Up to absolute constant factors, the deterministic conversion between the two models is $N \asymp nT$. In particular, the lower bound $N = \Omega(n \log n/\alpha^2)$ of Theorem E.3 is equivalent to a continuous-time lower bound $T = \Omega(\log n/\alpha^2)$.*

## H. A Technical Gap in the Analysis of Prior Work

We identify a gap in the analysis presented in (Tirukkonda et al., 2025). We have notified the authors, who are currently working to address it. Concretely, in the proof of Lemma 1 (step (c)), and again in the proof of Lemma 7 (Appendix B.5), the argument asserts that a certain conditional expectation of a ratio vanishes by invoking mean-zero and (marginal) independence of Gaussian noise terms. As we explain below, this cancellation is not justified when $(i,j) \in E$, because the denominator involves a future increment of coordinate $j$ which depends on earlier noise injected at coordinate $i$.

From a technical viewpoint, this is one major reason that our mixing-free analysis relies on update patterns of the form "$iiji$" rather than on a direct "$iji$" ratio argument. The additional update of $i$ is used to avoid exactly the type of dependence described above. In the separate mixing-based result, we are able to work with "$iji$" patterns because the analysis there is different and does not rely on this cancellation.

We now explain the gap in detail.

In Lemma 1, they consider an update pattern $n_1 < n_2 < n_3$ in which node $i$ is updated at times $n_1$ and $n_3$ and node $j$ is updated at time $n_2$, and they define a conditional expectation $\mathbb{E}_{\bar{x},c}[\cdot]$ that fixes the values of $X_{N(i)\setminus\{j\}}$ at the beginning of the window and conditions on the event $|X_j^{(n_2)} - X_j^{(n_1)}| \geq c$. In the proof, the step labeled (c) asserts that

$$\mathbb{E}_{\bar{x},c}\left[\frac{\varepsilon_i^{(n_3)} - \varepsilon_i^{(n_1)}}{X_j^{(n_2)} - X_j^{(n_1)}}\right] = 0,$$

citing the mean-zero and (marginal) independence of the Gaussian noise variables $\{\varepsilon_i^{(t)}\}$. A formally identical cancellation is used later in Appendix B.5 (in the proof of Lemma 7) when rewriting their test statistic as a signal term plus a ratio involving $\Delta\varepsilon_i/\Delta X_j$ and dropping the latter in conditional expectation.

**Why the denominator depends on earlier noise when $(i,j) \in E$.** Fix the "$iji$" update pattern $n_1 < n_2 < n_3$ from Lemma 1 of (Tirukkonda et al., 2025), where $i$ is updated at times $n_1$ and $n_3$ and $j$ is updated at time $n_2$, and enforce their accompanying event that no neighbor of $i$ or $j$ (other than possibly each other) updates inside the window.

Assume $(i,j) \in E$, so $\Theta_{ji} \neq 0$. Recall the Gaussian single-site update rule: when $u$ is updated at time $t$,

$$X_u^{(t)} = -\sum_{k \in N(u)} \frac{\Theta_{uk}}{\Theta_{uu}} X_k^{(t-1)} + \varepsilon_u^{(t)},$$

$$\varepsilon_u^{(t)} \sim \mathcal{N}\left(0, \frac{1}{\Theta_{uu}}\right),$$

Consider the randomness only through the lens of the single noise term $\varepsilon_i^{(n_1)}$ by fixing the pre-$n_1$ state, the update indices, and all other Gaussian noises $\{\varepsilon_u^{(t)} : (u,t) \neq (i,n_1)\}$. Under this fixing, the update at time $n_1$ gives an affine representation

$$X_i^{(n_1)} = m_i + \varepsilon_i^{(n_1)}$$

for a constant $m_i$ determined by what has been fixed. Since $i$ is not updated between $n_1$ and $n_2$, the update of $j$ at time $n_2$ uses $X_i^{(n_1)}$, and therefore

$$X_j^{(n_2)} = -\frac{\Theta_{ji}}{\Theta_{jj}} X_i^{(n_1)} - \sum_{k \in N(j) \setminus \{i\}} \frac{\Theta_{jk}}{\Theta_{jj}} X_k^{(n_2-1)} + \varepsilon_j^{(n_2)}$$

$$= m_j' - \frac{\Theta_{ji}}{\Theta_{jj}} \varepsilon_i^{(n_1)} + \varepsilon_j^{(n_2)},$$

for a constant $m_j'$ determined by the same fixing. Consequently, the increment appearing in their denominator satisfies

$$X_j^{(n_2)} - X_j^{(n_1)} = a + b\,\varepsilon_i^{(n_1)} + \varepsilon_j^{(n_2)},$$

$$b = -\frac{\Theta_{ji}}{\Theta_{jj}} \neq 0,$$

for a constant $a$ determined by the fixed past. In particular, when $(i,j) \in E$, the denominator is a function of $\varepsilon_i^{(n_1)}$, so it is not independent of the numerator noise term $\varepsilon_i^{(n_1)}$.

**The conditioning $|X_j^{(n_2)} - X_j^{(n_1)}| \geq c$ does not repair the issue.** In Lemma 1 (and again in Appendix B.5), (Tirukkonda et al., 2025) conditions on the event

$$\left| X_j^{(n_2)} - X_j^{(n_1)} \right| \geq c.$$

Under the affine form above, this event is exactly

$$\left| a + b\,\varepsilon_i^{(n_1)} + \varepsilon_j^{(n_2)} \right| \geq c,$$

which depends on $\varepsilon_i^{(n_1)}$. Thus, even after imposing the "$\geq c$" condition, the ratio term in step (c) involves a numerator noise component that is statistically coupled to the denominator.

**The ratio noise term need not have zero (conditional) expectation.** As a result, the cancellation invoked in step (c) would require showing that an expression of the form

$$\mathbb{E}\left[ \frac{\varepsilon_i^{(n_1)}}{a + b\,\varepsilon_i^{(n_1)} + \varepsilon_j^{(n_2)}} \;\middle|\; \left| a + b\,\varepsilon_i^{(n_1)} + \varepsilon_j^{(n_2)} \right| \geq c \right] = 0$$

holds under the relevant conditioning. There is no general reason for this to be true.

In fact, for $a = 0$ and $b = 1$, we have

$$\mathbb{E}\left[ \frac{\varepsilon_i^{(n_1)}}{\varepsilon_i^{(n_1)} + \varepsilon_j^{(n_2)}} \;\middle|\; \left| \varepsilon_i^{(n_1)} + \varepsilon_j^{(n_2)} \right| \geq c \right] = \frac{1}{2}$$

by symmetry in $\varepsilon_i^{(n_1)}$ and $\varepsilon_j^{(n_2)}$, as they are i.i.d.

Therefore, the vanishing of the ratio noise term cannot be justified solely from mean-zero and marginal independence of Gaussian noises.

**Implications and connection to our approach.** The discussion above shows that the specific cancellation used in step (c) of Lemma 1 of (Tirukkonda et al., 2025) is not justified as stated. The denominator contains the future increment $X_j^{(n_2)} - X_j^{(n_1)}$, which, when $(i,j) \in E$, can depend on the earlier noise $\varepsilon_i^{(n_1)}$. Consequently, the relevant ratio term need not have zero conditional expectation. Since the same cancellation is used again in Appendix B.5, in the proof of Lemma 7, the same issue propagates to the later separation argument built on that identity.

For our purposes, this explains why the mixing-free part of our analysis is based on $iiji$ patterns rather than on this direct $iji$ ratio argument. The additional update of $i$ removes the dependence created by the first $i$-update before the later update of $j$ enters the statistic, so the conditional-independence step needed in our proofs becomes valid. By contrast, when mixing is available, we also give a separate algorithm based on $iji$ patterns; that argument uses the mixing assumption and does not rely on the cancellation above.

## I. Non-Degeneracy Does Not Control Eigenvalues

In this appendix we demonstrate that the reciprocal of the minimum eigenvalue of the normalized precision matrix $\Theta'$ can be arbitrarily large even when all nonzero entries are bounded away from 0. Further, we show that this phenomenon is independent of the sparsity constraint: even when the graph of $\Theta'$ is $d$-sparse for $d > 2$, the edge-strength parameter $\alpha$ does not control $1/\lambda_{\min}(\Theta')$. Our construction is similar to Example (5) in (Misra et al., 2020).

**Proposition I.1.** *Fix $n \geq 2$, $\alpha < 1$, and $d \geq 2$. For any large number $N \in \mathbb{R}^+$, there exists a matrix $\Theta' \in \mathbb{R}^{n \times n}$ with minimum edge strength $\alpha$ and sparsity $d$, while $\lambda_{\min}^{-1} > N$.*

*Proof.* Consider the matrix

$$B_{\varepsilon} := \begin{bmatrix} 1 & 1 - \varepsilon \\ 1 - \varepsilon & 1 \end{bmatrix}$$

This matrix has minimum edge strength $1 - \varepsilon$.

The matrix $B_{\varepsilon}$ has eigenvectors $v_1 = [1, 1]$ and $v_2 = [1, -1]$, satisfying $B_{\varepsilon} v_1 = (2 - \varepsilon) v_1$ and $B_{\varepsilon} v_2 = \varepsilon v_2$. Hence $\lambda_{\min}^{-1} = 1/\varepsilon$. This means that for all $\alpha < 1$ and any large enough number $M > N$ with $M > \frac{1}{1-\alpha}$, the matrix $B_{1/M}$ has entries lower bounded by $1 - \frac{1}{M} > \alpha$, while the reciprocal of the minimum eigenvalue is $M$.

We can further extend the example of $B_{\varepsilon}$ to higher dimension to demonstrate that the edge-strength condition, together with sparsity constraints, can still allow large $\lambda_{\min}^{-1}$. For any $n > 2$, the block matrix

$$B_{n,\varepsilon} := \begin{bmatrix} B_{\varepsilon} & 0 \\ 0 & I_{n-2} \end{bmatrix},$$

where $I_{n-2}$ denotes the identity matrix in $n - 2$ dimensions, has minimum entry $1 - \varepsilon$ and sparsity $d = 2$. The eigenvalues of $B_{n,\varepsilon}$ are determined by its block components, so its minimum eigenvalue is $\varepsilon$ and its maximum eigenvalue is $2 - \varepsilon$; thus it has $\lambda_{\min}^{-1} = \frac{1}{\varepsilon}$. By the same argument as above, for any $\alpha < 1$ and $d > 2$, we may choose $M > \frac{1}{1-\alpha}$ and $M > N$, and then $B_{n,1/M}$ has $\lambda_{\min}^{-1}$ at least $M$ while having $\alpha$ edge strength and $d$-sparsity. $\square$

