# OpenReview forum: "Learning Gaussian Graphical Models from a Glauber Trajectory Without Mixing"
_ICML.cc/2026/Conference — ICML 2026 regular_

### Official Review · Reviewer_oMdU · 2026-03-07

**Soundness:** 2
**Presentation:** 2
**Significance:** 2
**Originality:** 2
**Overall Recommendation:** 2
**Confidence:** 3

**Summary:**

This paper studies structure learning for sparse Gaussian graphical models from a single continuous-time Glauber trajectory, rather than from i.i.d. samples. The main claim is a polynomial-time structure-learning algorithm that avoids any mixing-time assumption, under sparsity and a minimum normalized edge-strength condition. The method has three stages: estimate diagonal entries of the precision matrix, normalize the trajectory to an approximately unit-diagonal model, and estimate pairwise interactions from short update windows using robust median-based aggregation. The paper also gives a parameter-learning extension and an information-theoretic lower bound. A central technical point is that the manuscript identifies a dependence issue in prior work’s “iji” statistic and replaces it with an “iiji” update pattern to recover usable local edge information from a single correlated trajectory.

**Compliance With Llm Reviewing Policy:**

Affirmed.

**Final Justification:**

I am rejecting this paper because the rebuttal largely confirms, rather than resolves, my main concerns. My review questioned the realism of the observation model, the absence of any empirical validation, the limited extension beyond exact Glauber dynamics, the large constants, and the need for a clear limitations discussion. In response, the authors clarify that
- the theorem is only for exact continuous-time single-site Glauber dynamics with observed updates;
- they do not claim practical competitiveness;
- the large constants are tied to the present estimator and not merely loose analysis.

Because these remaining issues go to the paper’s scope, practical relevance, and generality, addressing them would require substantial additions to the paper, such as new experiments, sharper analysis, or new theory for broader observation models, rather than a short rebuttal. The paper’s current guarantees are also explicitly tied to a very specific setting in the main manuscript, namely exact continuous-time Glauber dynamics with observed updates, while the resulting sample complexity constants are extremely large.

**Key Questions For Authors:**

1. What realistic data-collection setting exactly matches your observation model? The current paper motivates the problem with broad temporally correlated applications, but the theorem assumes exact continuous-time single-site Glauber dynamics with observed updates.
2. Can you provide even a minimal synthetic evaluation? For example, recovery probability versus trajectory length, degree, and edge strength, plus a comparison to a thinning-based baseline or the nearest prior method where possible.
3. How far does the core idea extend beyond exact Markovian Glauber dynamics? In particular, is the “iiji” dependence fix specific to this model, or do you expect an analogous argument for other MCMC trajectories or partially observed trajectories?
4. Which step is really responsible for the large constants and the $\alpha^{-5}$ dependence?

**Limitations:**

No. I did not find an adequate discussion of limitations or potential negative societal impact in the visible manuscript. The paper should add a short, explicit limitations section stating that the theory applies only to exact continuous-time Markovian Glauber dynamics with observed updates, and that the guarantees may fail under partial observability, discretization, model mismatch, or other trajectory models from advanced Monte Carlo methods.

**Strengths And Weaknesses:**

Strength:
- Clear and well-motivated problem: learning sparse GGMs from a single Glauber trajectory without relying on mixing.
- The method has a coherent structure: diagonal estimation, normalization, local edge testing, and robust aggregation.
- The theory is reasonably complete, including structure recovery, parameter recovery, and a lower bound.

---

Weakness:
- The assumptions are mathematically clean but highly stylized, and the paper does not justify under what scenarios these assumptions hold.
- The result seems far from practically meaningful because the explicit constants (in Theorem C.1) are extremely large and the theory is not sharp.
- The paper lacks figures or summary tables to compare with previous works, making it hard to posit this work.
- There is no empirical validation at all: no simulations, no baselines, and no finite-sample evidence.

---

> ### Author Rebuttal · Authors · 2026-03-31
>
> Thank you for the thoughtful review. The main contribution is a polynomial-time structure-learning result for sparse GGMs from a single Glauber trajectory without any mixing-time assumption and without the extra regularity assumptions used in the only directly related prior work. This is not just a cleaner presentation of an existing argument: the paper identifies a serious gap in the prior `iji` analysis and replaces it with the correct `iiji` statistic.
>
> > The assumptions are mathematically clean but highly stylized ... What realistic data-collection setting exactly matches your observation model?
>
> The theorem is stated for exact continuous-time single-site Glauber dynamics with observed updates. A direct exact instance is Gibbs sampling for a Gaussian target, where update times, updated coordinates, and states are recorded. This is the cleanest local-update model for the question we study. The broader examples in the introduction are meant to motivate why temporally dependent, non-i.i.d. observations matter, not to suggest that every such application literally fits the theorem statement. Even for this model, the no-mixing question was **not** previously resolved under these assumptions. Solving that case is therefore already a meaningful contribution.
>
> > There is no empirical validation at all ... Can you provide even a minimal synthetic evaluation?
>
> This is a theory paper. Its claim is an existence theorem: polynomial-time structure learning is possible from a single trajectory without mixing and without extra regularity assumptions beyond sparsity and minimum normalized edge strength. We do not claim that the current estimator is practically competitive, and we do not want to suggest otherwise. For a paper of this kind, the main evaluation is whether the theorem is correct and whether it improves the assumptions and analysis over prior work. We will make that scope more explicit in the revision. See our responses to the other reviewers for more details on practicality.
>
> > The result seems far from practically meaningful because the explicit constants (in Theorem C.1) are extremely large and the theory is not sharp. Which step is really responsible for the large constants and the \( \alpha \) dependence?
>
> The blow-up comes from one concrete tradeoff. To keep hidden neighbor updates rare enough, the proof has to work on very short windows, with \(T=\alpha/(257d)\). But once the window is that short, a clean `iiji` sample is rare, because it requires four specific updates in prescribed subintervals. That rarity costs about a \(1/T^3\) factor in total observation time, and that is where most of the constant comes from. So the constant is not just an artifact of loose bookkeeping; it is tied to the current estimator.
>
> > How far does the core idea extend beyond exact Markovian Glauber dynamics? In particular, is the `iiji` dependence fix specific to this model, or do you expect an analogous argument for other MCMC trajectories or partially observed trajectories?
>
> The `iiji` fix is specific to the single-site local-update setting. For nearby models such as subsampling or small-block Gibbs, the same short-window viewpoint may still be relevant, and we expect it to extend to those settings. For systematic-scan Gibbs, Langevin dynamics, or other partially observed models, one is simply asking a different problem, so we would not expect the same theorem or the same statistic verbatim. We can add a short discussion clarifying that boundary.
>
> > The paper lacks figures or summary tables to compare with previous works, making it hard to position this work.
>
> There is essentially one directly comparable prior paper here. The comparison is therefore not a benchmark table. That paper assumes more and its key `iji` step does not go through as stated, while our paper removes the extra assumptions and uses a correct `iiji` statistic. We already discuss that work in detail. Since there is essentially one directly comparable prior paper, we do not think a separate comparison table would be informative here.
>
> > The paper should add a short, explicit limitations section stating that the theory applies only to exact continuous-time Markovian Glauber dynamics with observed updates ...
>
> The theorem already states its scope precisely. For readability, we can add a paragraph making explicit that the guarantee is for exact continuous-time single-site Glauber dynamics with observed updates. But we do not think the absence of a separate limitations section changes the technical contribution.
>
> Overall, the right lens for this paper is not whether it already solves every practical variant of the problem; we never claimed that. The question we study is the foundational mathematical one, and there the contribution is significant: we obtain the first correct polynomial-time no-mixing result for sparse GGMs from a single Glauber trajectory, and we explain exactly why the prior `iji` route fails and why `iiji` is needed instead.

---

> > ### Author Rebuttal · Reviewer_oMdU · 2026-04-03
> >
> > Thank the authors for the response. It clarified the paper’s scope and the source of the large constants. However, my main concerns remain. The theorem still applies only to exact continuous-time single-site Glauber dynamics with observed updates. The authors do not state the practicality of the current estimator, so it is hard to digest the significance of this work in real applications. These are core issues, so I do not think they can be fully resolved in a short rebuttal.

---

> > > ### Author Response · Authors · 2026-04-04
> > >
> > > Thank you for the acknowledgement. We understand the remaining concern. At this point, however, the remaining disagreement seems to concern scope and how theory papers should be evaluated, rather than technical correctness.
> > >
> > > Our claim is not that the current estimator is practically competitive, nor that every temporally dependent dataset literally matches the theorem statement. The claim is narrower and mathematical: in the exact single-trajectory Glauber setting, one can recover the structure in polynomial time without any mixing-time assumption and without the extra regularity assumptions used in the prior directly related work. In addition, the paper identifies a concrete technical gap in the prior `iji` analysis and replaces it with the correct `iiji` statistic.
> > >
> > > So while we understand the reviewer's preference for a broader or more practically oriented contribution, we believe the remaining concern is about scope rather than the validity or novelty of the result. We are happy to make the scope and the non-practicality of the current estimator even more explicit in the final version, but we believe the paper should be evaluated as a substantive theoretical contribution on its own terms.

---

### Official Review · Reviewer_aX2Y · 2026-03-09

**Soundness:** 2
**Presentation:** 1
**Significance:** 3
**Originality:** 3
**Overall Recommendation:** 2
**Confidence:** 4

**Summary:**

This work analyzes the problem of learning Gaussian Graphical Models (GGM) from a single trajectory of the associated Glauber dynamics, the goal being to infer the underlying precision matrix. The authors then construct informative local statistics using the updates of the local Markov chain in order to infer the entries of the matrix. These statistics are then analyzed using techniques from robust learning and demonstrate efficient learning despite noisy updates.

**Compliance With Llm Reviewing Policy:**

Affirmed.

**Final Justification:**

My primary concerns around this paper deal with presentation and exposition of the proof. The problem the authors attempt to solve is important to the statistical learning community, yet in its current shape, I do not recommend acceptance. As noted in my earlier comments, much of the actual content in the main body of the work is redundant, with the intuition behind the `iiji` updates explained twice. Moreover, the key engine behind the proof, robust statistics, is only given a scant mention, with no intuition provided behind why the diagonal needs to be whitened.

Likewise, as mentioned in my earlier comments, the notation is not standardized throughout and this makes the work difficult at times to understand. The main paper needs to be expounded on to ensure all aspects of the proof (including the claimed lower bound) are properly incorporated. The lack of practical application or experimental evidence does not effect my reading and opinion of the work, as it is a theory work.

With all this in mind, I keep my score. For the next iteration of this work, I would encourage the authors to clean up the presentation of the work, to make the techniques and main results easier to understand.

**Key Questions For Authors:**

My questions for this work can be found in the strengths and weaknesses section.

**Limitations:**

yes

**Strengths And Weaknesses:**

**Strengths**
> - The problem the paper seeks to solve is interesting and of interest to the statistical learning community.
> - The application of local statistics, e.g. "iiji", to learn the underlying matrix is original to this setting, and brings the assumptions over the model in line with the literature of learning Markov Random Fields from Glauber trajectories [2].

**Weaknesses**
> - The main body of the paper itself is structured in a somewhat unintuitive manner, with much space dedicated to the approach of [1] and insufficient space dedicated to the intuition behind the primary estimation algorithm. Until a detailed reading of the appendix, it was unclear how results from robust statistics were used, or why the authors were able to utilize "ii" or "iiji" updates.
> - The main results are presented in a somewhat confusing way, where it is unclear the *length* of the Glauber trajectory the authors are referring to denotes the number of single site updates of the dynamics or the interval of time the trajectory takes.
> - This uncertainty creeps in with respect to the claimed lower bound. The authors state estimation is possible using a length $\Omega(d^3 \log(n/\delta)/(\epsilon^5 \alpha^5))$ (Theorem 1.3, 1.4) trajectory, but also state information theoretically there exists a trajectory of length $O(n\log(nd/\delta)/\alpha^2)$ (Theorem D.1) where structure learning is impossible. The latter is normally much larger than the former as $n \gg d,\alpha$. This leads to an apparent contradiction.
> - The authors normalize the Glauber dynamics to have diagonal 1, and it is unclear from the main body and appendix what benefits this leads to in the estimation algorithm.
> - The choice of using "iiji" updates instead of "iji" updates is novel but is discussed twice in the main body of the paper alone, seemingly redundantly.
> - The presentation is a bit confusing to follow with the technical overview reading more akin to justification of using "iiji" updates adjoined to the steps of the algorithm with little intuition to go alongside it. The reasoning behind using robustness could be better highlighted in the main draft. It would be easier to understand the appendix, if there was extended discussion about the intermediate lemmas that lead to the main theorem, in lieu of overview paragraphs, which makes the draft difficult at times to read. Some of the lemmas are phrased in an awkward way, e.g. the difference between Lemma B.2 and B.7.
>- Detailed discussion about the algorithm for structured learning, and later parameter estimation is not given in the main text.


[1] Tirukkonda, V., Rayas, A., and Dasarathy, G. Structure learning in gaussian graphical models from glauber dynamics. In 2025 IEEE International Symposium on Information Theory (ISIT), pp. 1–6. IEEE, 2025.

[2] Gaitonde, J., Moitra, A., and Mossel, E. Bypassing the noisy parity barrier: Learning higher-order markov random fields from dynamics. In Proceedings of the 57th Annual ACM Symposium on Theory of Computing, pp. 348–359, 2025.

---

> ### Author Rebuttal · Authors · 2026-03-31
>
> Thank you for the careful reading. The main issues raised here are presentation issues, not technical flaws, and we can fix them directly in the revision.
>
> > The main body of the paper itself is structured in a somewhat unintuitive manner ... insufficient space dedicated to the intuition behind the primary estimation algorithm ... unclear how robust statistics were used, or why the authors were able to utilize `ii` or `iiji` updates.
>
> The extended discussion of [1] is deliberate: identifying the gap in [1] and explaining why `iji` is insufficient is part of our contribution. This is also part of the reason we have to use `iiji`. We will rebalance the exposition so the three-step algorithm is explicit in the main body: (i) use `ii` windows to estimate the diagonal via a robust variance estimator, (ii) normalize to the unit-diagonal setting, and (iii) use `iiji` windows to estimate pairwise interactions. The role of robustness is simple and should have been stated more directly: the update pattern is observable, but hidden neighbor updates inside the same window are not, so the selected windows are a mixture of clean and corrupted samples; this is exactly why median-based aggregation appears in both the diagonal and off-diagonal steps.
>
> > The main results are presented in a somewhat confusing way ... unclear whether the length of the Glauber trajectory denotes the number of single-site updates or the interval of time ... this leads to an apparent contradiction with the lower bound.
>
> There is no contradiction. Theorems 1.3 and 1.4 are stated in terms of the continuous-time horizon $T$, while Theorem D.1 is stated in terms of the number $N$ of single-site updates. These two viewpoints are equivalent for our purposes. In continuous-time Glauber dynamics, each coordinate has an independent Poisson clock of rate $1$, so by time $T$ the chain has made Poisson$(nT)$ updates. Standard concentration therefore lets one pass between bounds in $N$ and bounds in $T$ with no loss in the asymptotic rates. Thus
> $$
> N = \Omega\left(\frac{n \log(nd)}{\alpha^2}\right)
> \quad\Longleftrightarrow\quad
> T = \Omega\left(\frac{\log(nd)}{\alpha^2}\right)
> $$
> at the level of our bounds. We will make this conversion explicit and standardize notation throughout.
>
> > The authors normalize the Glauber dynamics to have diagonal $1$, and it is unclear what benefits this leads to in the estimation algorithm.
>
> Normalization is not required for structure learning, but it makes both the estimator and the analysis cleaner. After rescaling coordinate i by the square root of Theta_ii, the transformed process is still a Glauber trajectory, and the interaction parameter becomes Theta_ij divided by the square root of Theta_ii Theta_jj, which is exactly the normalized edge quantity in our assumptions. One could avoid normalization and instead estimate Theta_ij / Theta_ii and Theta_ij / Theta_jj directly, then combine them, but the argument becomes less clean. We also need diagonal estimation anyway for parameter learning, and the normalization step is doing real work there and is not merely a convenience like structure learning.
>
> > The choice of using `iiji` updates instead of `iji` updates is novel but is discussed twice in the main body ... seemingly redundantly.
>
> These two occurrences serve different purposes. The earlier discussion is relative to [1]: it explains why `iji` fails and why a different statistic is needed. The Technical Overview is where `iiji` is actually constructed and analyzed.
>
> > The presentation is a bit confusing ... the reasoning behind using robustness could be better highlighted ... the difference between Lemma B.2 and B.7.
>
> We will add a short roadmap to the appendix, state the robustness argument explicitly in the overview, and clarify that B.7 is just the all-coordinate version of the single-coordinate diagonal-estimation lemma obtained from B.2 by a union bound. This is a presentation issue, and easy to fix.
>
> > Detailed discussion about the algorithm for structure learning, and later parameter estimation is not given in the main text.
>
> We agree. In the revision we will move the pseudocode and a fuller description of both algorithms into the main body.
>
> Overall, we appreciate these comments and will revise the paper so that the estimator, the role of robustness, and the relation between the main bounds are much easier to follow.

---

> > ### Author Rebuttal · Reviewer_aX2Y · 2026-04-02
> >
> > I keep my score due to the lingering concerns from the rebuttal as expressed in my comment.
> > Although the underlying idea is important and interesting to the community of ICML, the presentation makes it difficult to understand the result itself and how the proof is structured. It would be nice if the paper consolidated redundant information and standardized notation throughout. The main body could have held an extended discussion of the techniques used in the proof.

---

> > > ### Author Response · Authors · 2026-04-04
> > >
> > > Thank you for the follow-up. We agree that the notation mismatch between the Technical Overview and the later formal analysis should be fixed; that is a straightforward correction.
> > >
> > > We also note that the acknowledgement says there are follow-up questions, but it does not actually state a specific follow-up question for us to answer. If there is a specific follow-up point the reviewer would like clarified, we would be happy to address it directly here.
> > >
> > > On the broader point, we respectfully disagree that the main paper does not discuss the techniques used in the proof. Section 1.1 explicitly directs the reader to Section 2 for a technical overview, and Section 2 then explains the estimator in the main paper: `ii` windows for diagonal estimation, the corruption viewpoint and robust variance estimation, normalization, `iiji` windows for off-diagonal estimation, and median aggregation under temporal dependence.
> > >
> > > So we agree that the notation should be unified, but we do not think it is accurate to say that the main paper does not discuss the techniques used in the proof; Section 2 was written for exactly that purpose.

---

### Official Review · Reviewer_SYjx · 2026-03-10

**Soundness:** 3
**Presentation:** 3
**Significance:** 3
**Originality:** 3
**Overall Recommendation:** 3
**Confidence:** 2

**Summary:**

This paper studies the structure and parameter learning for a sparse Gaussian Graph Models (GMM), under circumstances where the data are not i.i.d., but instead from a single trajectory of continuous-time Glauber dynamics started from an arbitrary state. The key challenge is that, under the sparsity and minimum edge strength assumption, the chain's mixing time can be arbitrarily large, so that the standard  ``mix and subsample'' reductions to approximate the classical i.i.d. setting are not feasible.


The main contribution is a polynomial-time algorithm to learn the graph structure. Given a single-site Gibbs sampling trajectory of length T, it can recover the precision matrix(determining which edge exists) with high probability no less than $1-\delta$, without making any assumption towards mixing time. This paper provides a theorem showing that a trajectory with $\Omega(d^3 \log(n/\delta) / \alpha^5)$ length is sufficient to recover the structure. Moreover, it also presents that a length $\Omega(d^3 \log(n/\delta) / (\alpha^5 \varepsilon^5))$ is sufficient to estimate the parameter with relative error less than $\varepsilon$ in the precision matrix.

Technically, the proposed algorithm leverages short update-pattern intervals in the trajectory: firstly estimates the diagonal term in the precision matrix via ''ii''-type events and robust variance estimation based on median standard deviation, then normalises the process into a diagonal-1 model, and finally tests candidate edges using ''iiji''-type patterns instead of  ''iji''. Similar to the diagnoal case, the robust estimator median is also employed in the estimates of the mean, to handle corruption in some intervals and temporal dependence.

A further contribution of this work is identifying a technical gap in a closely related prior research regarding ''iji'' patterns. The work shows that the claimed cancellation of a conditional expectation of a ratio is not generally justified when an edge exists between i and j. Because there is a dependence between the denominator and the earlier injected noise, the paper substitutes''iiji''  patterns instead.

**Compliance With Llm Reviewing Policy:**

Affirmed.

**Final Justification:**

the authors’ own acknowledgement that the estimator is not practically viable keep me at a weak reject.

**Key Questions For Authors:**

1.Regarding the practical viability and lack of empirical validation
While I recognize and appreciate that this is primarily a theoretical contribution, the complete absence of empirical evaluation makes it difficult to assess the practical utility of the proposed method.

i.Could the authors provide some numerical simulations to demonstrate that the estimator behaves as expected?


2.Regarding the complexity bounds and constants:
While the circumvention of mixing time is a significant theoretical result, but the explicit form of trajectory length required in Theorem C.1 includes the constant factor(over 4 billion) and the $O(\alpha^{-5})$ dependence suggest that recovering even moderately weak edges would require impractically long observation windows. You also mentioned that the polynomial dependence on $d$ and $\alpha$ might not be tight

i.Do the auhtors believe these dependencies and constants are results of the worst-case analytical boundary techniques or are they fundamental limitations of this "iiji" pattern estimator? Have you derived a tighter bound since the submission?

ii.Have the authors empirically observed successful structure recovery with much shorter trajectories than the bound suggests?



3.Regarding the restrictive continuous-time assumption
The proposed method relies on the continuous-time Glauber dynamics, however, in many real word settings (neuroscience or finance), data is typically collected at discrete time intervals. So multiple single site updates may occur between consecutive measurements and intermediate states are unobserved.

i. How sensitive is the proposed approach to such discrete time subsampling?

**Limitations:**

yes

**Strengths And Weaknesses:**

Soundness

Technically, the paper is rigorous and well-supported. It leans heavily into theory, but its main arguments and the way it analyses the problem are very clear. Setting of the problem of interest is shown explicitly: Under ($\alpha,d$)-sparse GGM observed through a single trajectory of continuous-time single-site Gibbs sampling, and describes why standard reductions to i.i.d. setting are unavailable when mixing can be arbitrarily slow. Within this setting, the paper proposes a polynomial-time procedure and provides explicit trajectory length requirements for both structure and parameter estimation in Theorem 1.3 \& 1.4, further led to a full proof in Appendix C.

Methodologically, the algorithm they chose aligns well with the initial setting(single trajectory and without mixing). They use short update patterns to isolate pairwise interactions, including a normalisation step to control coordinate wise effects and use robust aggregation to deal with the impact of occasional ``corrupted'' windows and temporal dependence.

In the appendix, they have also listed proof to some key properties, like distribution of "iiji'' statistics in lemma C.6 and provides independent proof of robust estimator.

The assumption this work makes is reasonable, which only relies on sparsity and minimum edge strength to learn a graph's structure, and there is no additional assumption imposed on the regularity conditions of the precision matrix. So the mixing time could be arbitrarily large -- consistent with their goal of without mixing.



Presentation

Overall, this paper features a reasonable and logical structure, with a clear outline. Firstly, it effectively establishes "single trajectory with potentially arbitrarily slow mixing'' as the core problem. Section 1.1 then presents explicit results on complexity, success probability and feasibility, while Section 1.2 compares the work against a recent literature to clarify its unique contributions and highlight the differences. Finally, the Technical Overview provides the intuition first and smoothly transitions to detailed technical proofs.

The explanation around why standard i.i.d. methods fail and why mixing is unreliable is highly effective and well motivated.

The main results are presented early and upfront, allowing readers to quickly grasp the promised guarantees regarding structure recovery, parameter estimation, trajectory length, and polynomial-time execution.

The way Section 1.3 identifies a technical gap in prior analysis and links it to their adoption of the "iiji'' pattern is highly persuasive and easy to remember.


There appears to be a notational inconsistency between the Technical Overview and the later formal analysis regarding the "iiji'' statistic (the roles of i and j and the corresponding differences/noise terms). I may be missing a convention, but it would help to align the notation and state the intended identity consistently throughout.


Significance
The paper tackles the problem of learning graph structures from time-correlated (non-i.i.d.) data, a setting that is much closer to reality in many real-world scenarios.

It leans more towards advancing our theoretical understanding and fundamental capabilities, while its direct impact on real-world engineering applications remains relatively limited.

There are some potential extensions for other researchers to explore: generalising the current framework from Gaussian graph models to broader classes of graphical models; more realistic observational settings instead of exact continuous-time Glauber trajectories; a correction to current complexity(a tighter upper/lower bound or whether length dependence on $\alpha^{-5}$ and $d^{3}$ could be improved).

The scope of this work's impact is specialized on the specific setting ``sparse GGM and single trajectory of Glauber dynamics'', but it identifies a technical gap in the most recent and related work, would possibly be influenced in this specialized research area.

Even though the setting is specialized, the strategy of using short update intervals (specifically that 'iiji' patterns) to isolate conditional independences, combined with robust aggregation to tackle the temporal dependence is valuable for other researchers to concern when dealing with other dependent datasets.


Originality

This paper mainly presents two new viewpoints. Firstly, it extends "structure learning'' from standard i.i.d. methods to reliance on single continuous time trajectory without mixing. Secondly, it highlights the key role of the new ``iiji'' design in dealing with dependencies.

The paper introduces new task/setting: learning structure and parameter relying on single continuous Glauber dynamics trajectory without mixing. It adopts new short update interval(ii,iiji), and presents explicit upper bound of trajectory length under this setting, as well as a polynomial time feasibility.

This approach gives a novel combination of local short update window patterns with normalisation and robust estimation in a way that consistent with the scenarios of non i.i.d. single trajectory observations.

There is also an obvious difference between this work and related literatures with a relatively throughout justification:

Compared with the most close research, this paper shows the graph structure still could be learned under no additional regularity conditions except for sparsity and minimum edge strength assumption. Under less assumptions made, it identifies a technical gap in the previous research(inappropriate cancellation of in "iji'') and provides "iiji'' as correction.



Weakness

Although the paper makes solid theoretical contributions and provides rigorous bounds under its assumed scenarios, the paper is entirely theoretical and lacks any empirical evaluation. For demonstrating the algorithm's behaviour via numerical simulations, even on synthetic sparse GGMs, is crucial to help validate the proposed estimators behave as illustrated.

Without empirical validation, it is difficult to assess whether the proposed estimator is practically viable, how it behaves under finite-sample regimes, or how robust it is to minor wrong model specifications.

Such experiments would also make the work more accessible to a broader machine learning audience by connecting the theoretical window-based construction to observable performance in finite samples

---

> ### Author Rebuttal · Authors · 2026-03-31
>
> We thank the reviewer for the thoughtful and detailed review. We are glad the reviewer found the core theorem, the no-mixing setting, and the role of the `iiji` statistic compelling. The main issues raised are about scope, practicality, and how broadly the current proof should be interpreted. We address those points directly below.
>
> > There appears to be a notational inconsistency between the Technical Overview and the later formal analysis regarding the `iiji` statistic ...
>
> Thank you for catching this. The notation in the technical overview is not fully aligned with the later formal analysis. We have fixed this in the revision and will state the intended identity consistently throughout.
>
> > The paper is entirely theoretical and lacks empirical evaluation ... Could the authors provide numerical simulations? ... Have the authors empirically observed successful structure recovery with much shorter trajectories than the bound suggests?
>
> This paper is not making a practicality claim. The question we answer is whether one can recover the structure of a sparse GGM from a single Glauber trajectory in polynomial time, without any mixing-time assumption and without additional regularity assumptions beyond sparsity and minimum edge strength. That was an open problem, and the paper resolves it affirmatively.
>
> We are therefore not presenting the current estimator as practically competitive. In fact, from the experiments we have run, this does not look like a case where a pessimistic theorem is hiding a practical method: the present estimator seems genuinely impractical. So on this point we want to be direct. The contribution here is conceptual: identifying `iiji` as the right local statistic, showing why `iji` is insufficient, and proving polynomial-time learnability without mixing under these weak assumptions. We will make that scope much clearer in the revision.
>
> > The explicit trajectory length in Theorem C.1 includes a constant over $4$ billion ... are these dependencies artifacts of worst-case analysis, or limitations of the `iiji` estimator? Have you derived a tighter bound since submission?
>
> For the current estimator, the big constant and the impracticality come from the same source: the tradeoff between corruption rate and rate of occurrence. To keep the corruption rate within what the robust median estimator can tolerate, the local window length has to be very small; in our proof this is $T=\alpha/(257d)$. But once $T$ is that small, an informative `iiji` window becomes rare, since it requires several prescribed updates inside a window of length $T$. So the same choice that controls corruption also drives down the observation rate, and the trajectory length must blow up to compensate. Quantitatively, the pattern probability scales like $T^4$ while each candidate window only has length $T$, which creates a $1/T^3$ penalty in observation time even before concentration is applied. That is the main reason for both the huge constant and the poor polynomial dependence. In that sense this is mostly a limitation of the present estimator, not just proof bookkeeping. We have not derived a materially tighter theorem since submission.
>
> > The proposed method relies on continuous-time Glauber dynamics ... How sensitive is the approach to discrete-time subsampling?
>
> The current theorem is for the fully observed continuous-time single-site trajectory. The `iiji` statistic uses the exact order of a small number of single-site updates and the absence of neighbor updates inside the window. If one only observes snapshots at a coarse discrete time scale, that ordering information is lost: multiple hidden updates may occur between observations, and then the endpoints no longer tell us whether a clean `iiji` pattern occurred or how many confounding neighbor updates intervened. In that regime the present proof does not go through.
>
> That being said, we do expect a fine-sampling extension. If the observation mesh is fine enough that, with high probability, each observation interval contains at most one hidden update, then the underlying update sequence can essentially be reconstructed from the sampled path, and the same local-window argument should survive with an additional sampling-overhead factor. But coarse subsampling is not a small perturbation of our setting; it removes exactly the local ordering information the estimator is built around. We will state this limitation explicitly in the revision.
>
> We thank the reviewer again. In the revision we will (i) fix the `iiji` notation mismatch, (ii) make the theory-first scope explicit and avoid any suggestion of practical competitiveness, and (iii) clarify both the source of the large constants and how our method is or is not robust to subsampling.

---

> > ### Author Rebuttal · Reviewer_SYjx · 2026-04-03
> >
> > I appreciate the rigour and conceptual contribution, but the reliance on fully observed continuous-time single-site dynamics, the complete lack of empirical validation, and the authors’ own acknowledgement that the estimator is not practically viable keep me at a weak reject.

---

> > > ### Author Response · Authors · 2026-04-04
> > >
> > > Thank you for the clarification. We appreciate the acknowledgement that the paper’s technical contribution is rigorous and conceptually meaningful.
> > >
> > > We agree that the present submission has important limitations: it studies the exact fully observed continuous-time single-site Glauber model, it does not include empirical evaluation, and it does not claim practical competitiveness. We will make those limitations much more explicit in the revision.
> > >
> > > At the same time, this submission is a theory paper. Its goal is to resolve the no-mixing polynomial-time learnability question for sparse GGMs from a single Glauber trajectory under only sparsity and minimum normalized edge strength, and to identify and correct a technical gap in the closest prior work. In that sense, the main question for this paper is whether it makes a sound and meaningful theoretical advance.
> > >
> > > We therefore respectfully hope the paper will be evaluated primarily on that theoretical contribution, rather than on the absence of empirical validation or immediate practical competitiveness. We agree that more realistic observation models, experiments, and practically improved estimators are important next steps, but we do not believe that the absence of those extensions in the present submission diminishes the significance of the main theoretical result established here.

---

### Official Review · Reviewer_bVWb · 2026-03-17

**Soundness:** 3
**Presentation:** 3
**Significance:** 3
**Originality:** 3
**Overall Recommendation:** 5
**Confidence:** 3

**Summary:**

The authors study a relevant issue: learning the structure of sparse Gaussian Graphical Models (GGMs) from temporally dependent data, namely a single Glauber dynamics trajectory, rather than i.i.d. samples. They propose a polynomial-time algorithm that recovers the conditional independence graph without relying on mixing-time assumptions, under only sparsity and minimum edge-strength conditions. The method combines variance estimation and normalization, local edge tests using carefully designed update patterns, and robust median-based aggregation. It also provides guarantees for parameter estimation. Additionally, the paper identifies a technical flaw in prior work, which motivates its improved approach.

**Compliance With Llm Reviewing Policy:**

Affirmed.

**Final Justification:**

My (minor) questions are adressed. I am fine with the fact that this paper "is not making a practicality claim", but is rather identifying the right object to be at the heart of their new method, solving an open problem, which makes it a valuable contribution. I like this paper and maintain my positive assessment.

**Key Questions For Authors:**

p.6 "Note the events of (1) and (2) occurring are independent from each other, and as T → 0, the probability (2) occurs goes to zero." : but so is the probability of (1) to occur, no? Just found this point confusing.

p.2 "In paritcular" typo

p.3 "Under the condition that Θ has diagonal 1, these parameters are equivalent up to constants." : they are equal

p.6 second column, I do not see where the 0.6 comes from in \sqrt{2}/0.6. I guess it should be c ?

**Limitations:**

Adding numerical experiments or simulations would help validate the approach, illustrate robustness, and provide insight into practical performance.

Besides, I think more insights on the algorithms could be given in the main text.

**Strengths And Weaknesses:**

The work addresses a fundamental limitation of prior approaches by avoiding reliance on mixing-time assumptions. This is particularly important since mixing can be arbitrarily slow in GGMs.

The proposed method is an efficient (polynomial-time) algorithm for structure learning from a single Glauber trajectory under weak assumptions.

The use of carefully designed update patterns (notably the “iiji” pattern) to isolate pairwise interactions is elegant and resolves subtle dependence issues that break prior methods. The identification of a gap in Tirukkonda et al. (2025) strengthens the paper’s contribution and clarifies the need for new techniques.

---

> ### Author Rebuttal · Authors · 2026-03-31
>
> Thank you for the careful reading and the positive overall assessment. We are glad the reviewer found the no-mixing result, the `iiji` statistic, and the identification of the gap in prior work compelling. We address the points raised by the reviewer.
>
> ### Limitations
>
> > Adding numerical experiments or simulations would help validate the approach, illustrate robustness, and provide insight into practical performance.
>
> This is the one point where we want to be completely explicit about scope. This paper is not making a practicality claim. The question we address is whether one can recover the structure of a sparse GGM from a single Glauber trajectory in polynomial time **without** any mixing-time assumption and **without** additional regularity assumptions beyond sparsity and minimum edge strength. This was open problem, and the paper resolves it affirmatively.
>
> We are therefore **not** presenting the current estimator as practically competitive. The explicit constants already make that clear. More importantly, the bottleneck is not just a loose proof constant. To keep the corruption rate below what the robust estimator can tolerate, the analysis works with windows of length about $T=\alpha/(257d)$. A usable `iiji` sample then requires several prescribed updates inside such a window, so informative windows are intrinsically rare. This creates a real $1/T^3$ observation-time penalty before concentration is even applied, which is why the constants become so large.
>
> It is true that in some theory papers the constants are pessimistic and practice turns out much better. From the experiments we have performed, this does **not** appear to be one of those cases: the present estimator seems genuinely impractical. So on this point we want to be direct. Practical competitiveness is not the claim of the paper, and it would be misleading to present the current algorithm as if it were close to that regime.
>
> At the same time, the underlying local-window idea is not brittle. Since our algorithms are already based on robust estimators, our algorithms are already robust up to the corruption rates already considered in the paper. Beyond that threshold, if each actual update is revealed independently with probability $p$, then the same fixed-length `ii` and `iiji` windows should still be usable, at the cost of a polynomial factor in $1/p$ more intervals, since all designated updates in a window must now be observed.
>
> More broadly, this follows a very common pattern in theoretical computer science and statistics: first one identifies the right object, dependence structure, and proof mechanism and proves a polynomial-time theorem, often with constants that are not practically useful; only later does follow-up work simplify the estimator, sharpen the analysis, and sometimes make the method practical. That is how this paper should be read. The contribution here is conceptual: identifying `iiji` as the right local statistic, showing why `iji` is insufficient, and resolving the no-mixing possibility question under these weak assumptions.
>
> > Besides, I think more insights on the algorithms could be given in the main text.
>
> This is fair. In the revision, we will expand the discussion of the normalization step, make the role of the `ii` and `iiji` patterns more explicit, and clarify why robust median aggregation is the right tool once corrupted windows are unavoidable.
>
> ### Questions
>
> > p.6: "Note the events of (1) and (2) occurring are independent from each other, and as $T \to 0$, the probability (2) occurs goes to zero." But so is the probability of (1) to occur, no?
>
> This sentence has a typo. Event (1) is the observable sampling event, while corruption corresponds to the **failure** of (2), not the occurrence of (2). As $T \to 0$, $\Pr[(1)] \to 0$ while $\Pr[(2)] \to 1$. The actual point we use is independence, namely that conditioning on (1) does not bias the corruption rate. We will rewrite this paragraph to make the roles of the events clear.
>
> > p.2: "In paritcular" typo
>
> Thank you for catching this; we will fix it.
>
> > p.3: "Under the condition that $\Theta$ has diagonal 1, these parameters are equivalent up to constants." They are equal.
>
> Correct. In the unit-diagonal case they are exactly equal. We will fix this.
>
> > p.6 second column: I do not see where the $0.6$ comes from in $\sqrt{2}/0.6$. I guess it should be $c$?
>
> Yes, the generic expression is $\sqrt{2}/c$. In the technical overview we intended to set $c=0.6$, but we did not state this explicitly before writing $\sqrt{2}/0.6$. We will fix this by making the specialization explicit.

---

> > ### Author Rebuttal · Reviewer_bVWb · 2026-04-07
> >
> > My (minor) questions are adressed. I am fine with the fact that this paper "is not making a practicality claim", but is rather identifying the right object to be at the heart of their new method, solving an open problem, which makes it a valuable contribution. I like this paper and maintain my positive assessment.

---

> > > ### Author Response · Authors · 2026-04-07
> > >
> > > Thank you for the follow-up and for the positive assessment. We appreciate that our clarifications have addressed your concerns.

---

### Decision · Program_Chairs · 2026-04-30

**Decision:**

Accept (regular)

**Comment:**

This paper studies structure learning for sparse Gaussian graphical models from a single Glauber trajectory, without any mixing-time assumption. The main contribution is a polynomial-time recovery result in this setting, together with a new statistic that corrects a dependence issue in the closest prior work.

This is fundamentally a theory paper, and I think it should be evaluated as such. The main question here is not practical competitiveness, but whether the paper establishes a meaningful new theoretical result. On that axis, the reviewers broadly agree that the problem is interesting and that the paper contains a genuine conceptual contribution.

The negative reviews focus mainly on the narrowness of the model, the absence of experiments, the large constants, and the fact that the current estimator is not practically useful. These are real limitations, but for a theory paper I do not view them as decisive unless they point to a deeper issue with correctness or significance. After reading the reviews and rebuttal, I do not see evidence of a fatal technical flaw.

Overall, I find that the paper makes a meaningful theoretical advance: it resolves a natural no-mixing question in this model and introduces the right local statistic to do so. The weaknesses clearly limit enthusiasm, but they do not outweigh the contribution. I therefore recommend Accept.